# Context-aware deconvolution of cell–cell communication with Tensor-cell2cell

Erick Armingol [1,2,7], Hratch M. Baghdassarian [1,2,7], Cameron Martino [1,2,3], Araceli Perez-Lopez[4], Caitlin Aamodt[2], Rob Knight [2,3,5,6] & Nathan E. Lewis[2,6 ✉]

Cell interactions determine phenotypes, and intercellular communication is shaped by cellular contexts such as disease state, organismal life stage, and tissue microenvironment. Single-cell technologies measure the molecules mediating cell–cell communication, and emerging computational tools can exploit these data to decipher intercellular communication. However, current methods either disregard cellular context or rely on simple pairwise comparisons between samples, thus limiting the ability to decipher complex cell–cell communication across multiple time points, levels of disease severity, or spatial contexts. Here we present Tensor-cell2cell, an unsupervised method using tensor decomposition, which deciphers context-driven intercellular communication by simultaneously accounting for multiple stages, states, or locations of the cells. To do so, Tensor-cell2cell uncovers context-driven patterns of communication associated with different phenotypic states and determined by unique combinations of cell types and ligand-receptor pairs. As such, Tensor-cell2cell robustly improves upon and extends the analytical capabilities of existing tools. We show Tensor-cell2cell can identify multiple modules associated with distinct communication processes (e.g., participating cell–cell and ligand-receptor pairs) linked to severities of Coronavirus Disease 2019 and to Autism Spectrum Disorder. Thus, we introduce an effective and easy-to-use strategy for understanding complex communication patterns across diverse conditions.

[1] Bioinformatics and Systems Biology Graduate Program, University of California, San Diego, La Jolla, CA 92093, USA. [2] Department of Pediatrics, University of California, San Diego, La Jolla, CA 92093, USA. [3] Center for Microbiome Innovation, University of California San Diego, La Jolla, CA 92093, USA. [4] Biomedicine Research Unit, Facultad de Estudios Superiores Iztacala, Universidad Nacional Autónoma de México, Tlalnepantla, México 54090, México. [5] Department of Computer Science and Engineering, University of California San Diego, La Jolla, CA 92093, USA. [6] Department of Bioengineering, University of California, San Diego, La Jolla, CA 92093, USA. [7] These authors contributed equally: Erick Armingol, Hratch Baghdassarian. ✉email: nlewisres@ucsd.edu

Organismal phenotypes arise as cells adapt and coordinate their functions through cell–cell interactions within their microenvironments[1]. Variations in these interactions and the resulting phenotypes can occur because of genotypic differences (e.g., different subjects) or the transition from one biological state or condition to another[2] (e.g., from one life stage into another, migration from one location into another, and transition from health to disease states). These interactions are mediated by changes in the production of signals and receptors by the cells, causing changes in cell–cell communication (CCC). Thus, CCC is dependent on temporal, spatial and condition-specific contexts[3], which we refer to here as cellular contexts. "Cellular contexts" refer to variation in genotype, biological state or condition that can shape the microenvironment of a cell and therefore its CCC. Thus, CCC can be seen as a function of a context variable that is not necessarily binary and can encompass multiple levels (e.g., multiple time points, gradient of disease severities, different subjects, distinct tissues, etc.). Consequently, varying contexts trigger distinct strength and/or signaling activity[1,4–6] of communication, leading to complex dynamics (e.g., increasing, decreasing, pulsatile and oscillatory communication activities across contexts). Importantly, unique combinations of cell–cell and ligand-receptor (LR) pairs can follow different context-dependent dynamics, making CCC hard to decipher across multiple contexts.

Single-cell omics assays provide the necessary resolution to measure these cell–cell interactions and the ligand-receptor pairs mediating CCC. While computational methods for inferring CCC have been invaluable for discovering the cellular and molecular interactions underlying many biological processes, including organismal development and disease pathogenesis[5], current approaches cannot account for high variability in contexts (e.g., multiple time points or phenotypic states) simultaneously. Existing methods lose the correlation structure across contexts since they involve repeating analysis for each context separately, disregarding informative variation in CCC across such factors as disease severities, time points, subjects, or cellular locations[7]. Additional analysis steps are required to compare and compile results from pairwise comparisons[8–11], reducing the statistical power and hindering efforts to link phenotypes to CCC. Moreover, this roundabout process is computationally expensive, making analysis of large sample cohorts intractable. Thus, new methods are needed that analyze CCC while accounting for the correlation structure across multiple contexts simultaneously.

Tensor-based approaches such as Tensor Component Analysis[12] (TCA) can deconvolve patterns associated with the biological context of the system of interest. While matrix-based dimensionality reduction methods such as Principal Component Analysis (PCA), Non-negative Matrix Factorization (NMF), Uniform Manifold Approximation and Projection (UMAP) and t-distributed Stochastic Neighbor Embedding (t-SNE) can extract low-dimensional structures from the data and reflect important molecular signals[13,14], TCA is better suited to analyze multidimensional datasets obtained from multiple biological contexts or conditions[7] (e.g., time points, study subjects and body sites). Indeed, TCA outperforms matrix-based dimensionality reduction methods when recovering ground truth patterns associated with, for example, dynamic changes in microbial composition across multiple patients[15] and neuronal firing dynamics across multiple experimental trials[12]. TCA exhibits superior performance because it does not require the aggregation of datasets across varying contexts into a single matrix. It instead organizes the data as a tensor, the higher-order generalization of matrices, which better preserves the underlying context-driven correlation structure by retaining mathematical features that matrices lack[16,17]. Thus, with the correlation structure retained, the use of TCA with expression data across many contexts allows one to gain a detailed understanding of how context shapes communication, as well as the specific molecules and cells mediating these processes.

Here, we introduce Tensor-cell2cell, a TCA-based strategy that deconvolves intercellular communication across multiple contexts and uncovers modules, or latent context-dependent patterns, of CCC. These data-driven patterns reveal underlying communication changes given the simultaneous interaction between contexts, ligand-receptor pairs, and cells. We first show that Tensor-cell2cell successfully extracts temporal patterns from a simulated dataset. We also illustrate that Tensor-cell2cell is broadly applicable, enabling the study of diverse biological questions associated with COVID-19 severity and Autism Spectrum Disorder (ASD). While our approach can simultaneously analyze more than two samples, we show that Tensor-cell2cell is faster, demands less memory and can achieve better accuracy in separating context-specific information than simpler analyses accessible to other tools. We further demonstrate that Tensor-cell2cell can leverage existing CCC tools by using their output communication scores to analyze multiple contexts. Thus, Tensor-cell2cell's easily interpretable output leverages existing tools, and enables quick identification of key mediators of cell–cell communication across contexts, both reproducing known results and identifying previously unreported interactors.

## Results

**Deciphering context-driven communication patterns with Tensor-cell2cell.** Organizing biological data through a tensor preserves the underlying correlation structure of the biological conditions of interest[12,15,17]. Extending this approach to infer cell–cell communication enables analysis of important ligand-receptor pairs and cell–cell interactions in a context-aware manner. Accordingly, we developed Tensor-cell2cell, a method based on tensor decomposition[17] that extracts context-driven latent patterns of intercellular communication in an unsupervised manner. Briefly, Tensor-cell2cell first generates a 4D-communication tensor that contains non-negative scores to represent cell–cell communication across different conditions (Fig. 1a–c). Then, a non-negative TCA[18] is applied to deconvolve the latent CCC structure of this tensor into low-dimensional components or factors (Fig. 1d–e). Thus, each of these factors can be interpreted as a module or pattern of communication whose dynamics across contexts is indicated by the loadings in the context dimension (Fig. 1e).

To demonstrate how Tensor-cell2cell recovers latent patterns of communication, we simulated a system of 3 cell types interacting through 300 LR pairs across 12 contexts (represented in our simulation as time points) (Fig. 2a). We built a 4D-communication tensor that incorporates a set of embedded patterns of communication that were assigned to certain LR pairs used by specific pairs of interacting cells, and represented through oscillatory, pulsatile, exponential, and linear changes in communication scores (Fig. 2a–f; see Supplementary Notes for further details of simulating and decomposing this tensor). Using Tensor-cell2cell, we found that four factors led to the decomposition that best minimized error (Supplementary Fig. 1a), consistent with the number of introduced patterns (Fig. 2f). This was robustly observed in multiple independent simulations (Supplementary Fig. 2a).

Our simulation-based analysis further demonstrates that Tensor-cell2cell accurately detects context-dependent changes of communication, and identifies which LR pairs, sender cells, and receiver cells are important (Fig. 2g). In particular, the context loadings of the TCA on the simulated tensor accurately recapitulate the introduced patterns (Fig. 2f, g), while

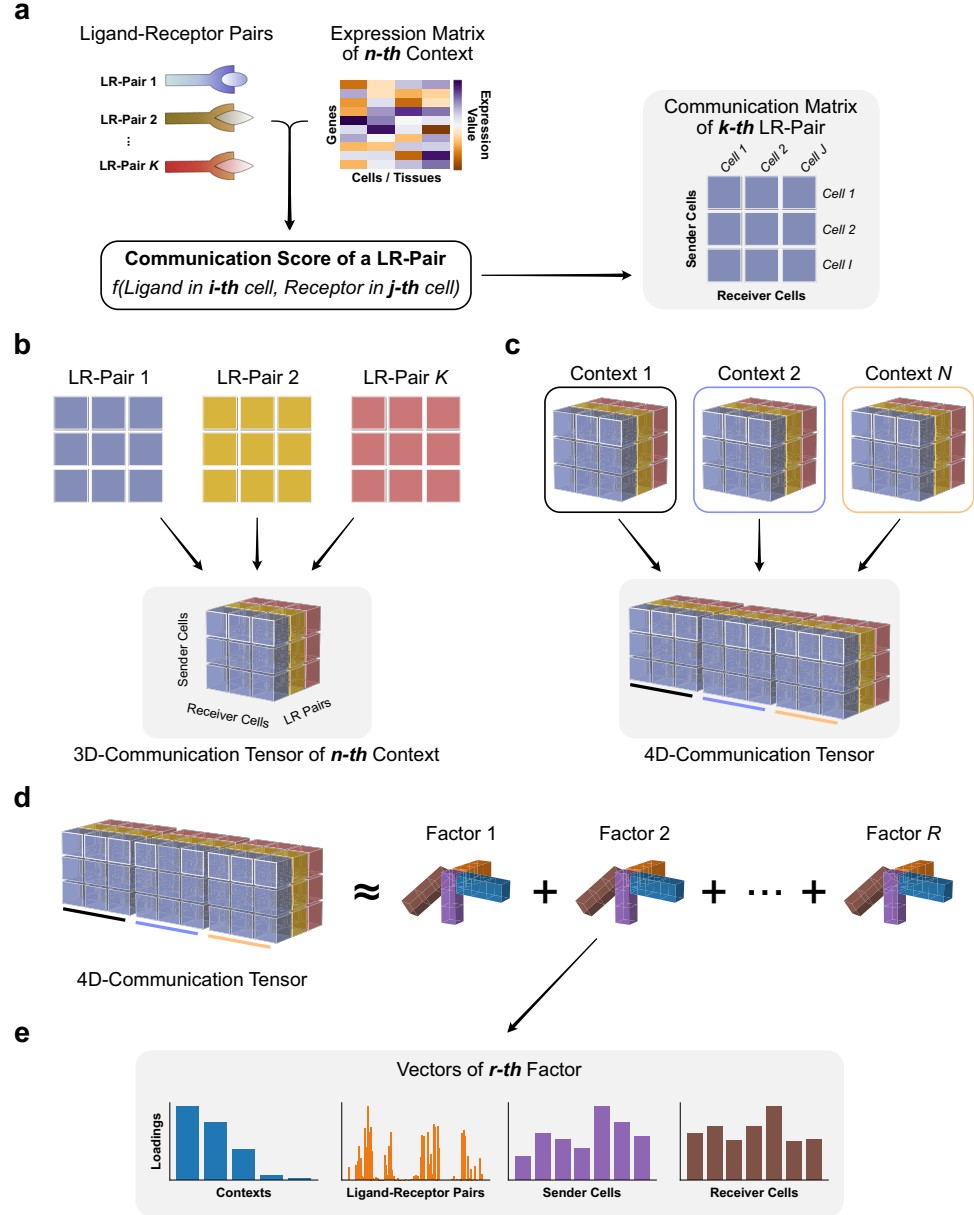

**Fig. 1 Tensor representation and factorization of cell–cell communication.** In a given context (*n*-th context among N total contexts), cell–cell communication scores (see available scoring functions in ref. 5) are computed from the expression of the ligand and the receptor in a LR pair (k-th pair among K pairs) for a specific sender-receiver cell pair (i-th and j-th cells among I and J cells, respectively). This results in a communication matrix containing all pairs of sender-receiver cells for that LR pair (**a**). The same process is repeated for every single LR pair in the input list of ligand-receptor interactions, resulting in a set of communication matrices that generate a 3D-communication tensor (**b**). 3D-communication tensors are built for all contexts and are used to generate a 4D-communication tensor wherein each dimension represents the contexts (colored lines), ligand-receptor pairs, sender cells and receiver cells (**c**). A non-negative TCA model approximates this tensor by a lower-rank tensor equivalent to the sum of multiple factors of rank-one (R factors in total) (**d**). Each component or factor (r-th factor) is built by the outer product of interconnected descriptors (vectors) that contain the loadings for describing the relative contribution that contexts, ligand-receptor pairs, sender cells and receiver cells have in the factor (**e**). For interpretability, the behavior that context loadings follow represent a communication pattern across contexts. Hence, the communication captured by a factor is more relevant or more likely to be occurring in contexts with higher loadings. Similarly, ligand-receptor pairs with higher loadings are the main mediators of that communication pattern. By constructing the tensor to account for directional interactions (panels a, b), ligands and receptors in LR pairs with high loadings are mainly produced by sender and receiver cells with high loadings, respectively.

ligand-receptor and cell loadings properly capture the ligand-receptor pairs, sender cells and receiver cells assigned as participants of the cognate pattern (Fig. 2g). Indeed, we observed a concordance between the "ground truth" LR pairs assigned to a pattern and their respective factor loadings through Jaccard index and Pearson correlation metrics (Supplementary Tables 1–2). Moreover, Tensor-cell2cell robustly

recovered communication patterns when we added noise to the simulated tensor (Supplementary Fig. 2 and Supplementary Notes).

**Tensor-cell2cell robustly extends cell–cell communication analysis.** To demonstrate the power of accounting for multiple contexts simultaneously, we compared the computational

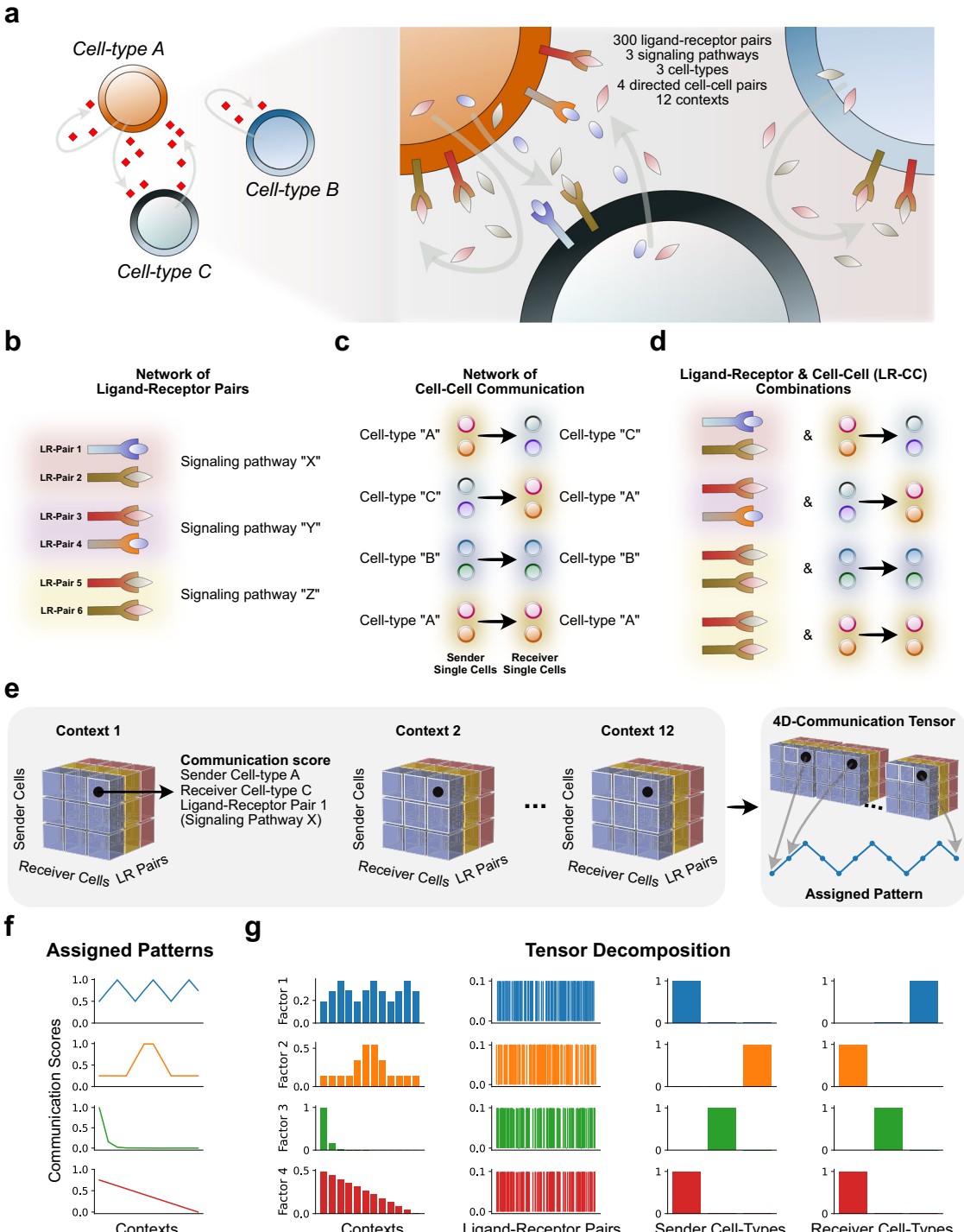

efficiency and accuracy of our method with respect to CellChat[10], the only tool that summarizes multiple pairwise comparisons in an automated manner (Table 1). Since CellChat cannot extract patterns of CCC across multiple contexts, we instead use the output of its joint manifold learning on pairwise-based changes in signaling pathways as a comparable proxy to the output of Tensor-cell2cell. Despite the use of these proxy comparisons, we emphasize that the conceptual outputs reported by Tensor-cell2cell are unique. Briefly, we found that Tensor-cell2cell is faster, uses less memory, and achieves higher accuracy when analyzing CCC of multiple samples (Supplementary Fig. 3); using a GPU further increases computational speed of Tensor-cell2cell. See more details regarding this comparison in the Methods and

*Tensor-cell2cell is fast and accurate* section of the Supplementary Notes.

A major advantage of Tensor-cell2cell is that it acts as a robust dimensionality reduction method for any communication scores arranged as a tensor. To illustrate this, we set out to harness the sample-wise communication scoring outputs of other tools. Tensor-cell2cell can restructure these outputs into a 4D-communication tensor (Fig. 1), extending their capabilities to recover context-dependent patterns of communication. This generalizability enables users to employ any scoring method. Thus, we ran Tensor-cell2cell on communication scores generated by sample-specific analysis with CellPhoneDB[19], CellChat[10], NATMI[9], and SingleCellSignalR[20], as well as the built-in scoring

**Fig. 2 Tensor-cell2cell recovers simulated communication patterns. a** Cell–cell communication scenario used for simulating patterns of communication across different contexts (here each a different time point). **b** Examples of specific ligand-receptor (LR) and (**c**) cell–cell pairs that participate in the simulated interactions. Individual LR pairs and cell pairs were categorized into groups of signaling pathways and cell types, respectively. In this simulation, signaling pathways did not overlap in their LR pairs, and each pathway was assigned 100 different LR pairs. **d** Distinct combinations of signaling pathways with sender-receiver cell type pairs were generated (LR-CC combinations). LR-CC combinations that were assigned the same signaling pathway overlap in the LR pairs but not in the interacting cell types. **e** A simulated 4D-communication tensor was built from each time point's 3D-communication tensor. Here, a communication score was assigned to each ligand-receptor and cell–cell member of a LR-CC combination. Each communication score varied across time points according to a specific pattern. **f** Four different patterns of communication scores were introduced to the simulated tensor by assigning a unique pattern to a specific LR-CC combination. From top to bottom, these patterns were an oscillation, a pulse, an exponential decay and a linear decrease. The average communication score (y-axis) is shown across time points (x-axis). This average was computed from the scores assigned to every ligand-receptor and cell–cell pair in the same LR-CC combination. **g** Results of running Tensor-cell2cell on the simulated tensor. Each row represents a factor, and each column a tensor dimension, wherein each bar represents an element of that dimension (e.g., a time point, a ligand-receptor pair, a sender cell or a receiver cell). Factor loadings (y-axis) are displayed for each element of a given dimension. Here, the factors were visually matched to the corresponding latent pattern in the tensor, and their loadings were normalized to unit Euclidean length. Assigned pattern scores and loading source data are provided in the Source Data file.

**Table 1 Methodological strategy and context-based analysis in available tools.**

| Tool | Communication Score[a] | Context Evaluation | Simultaneous Contexts | Multimeric LR pairs | Data Resolution | Platform | Refs. |
|---|---|---|---|---|---|---|---|
| Tensor-cell2cell | Expression Mean, Expression Product and Geometric Mean | Builds a tensor with all contexts simultaneously and runs a tensor decomposition, accounting for the correlation structure across contexts | Unlimited[b] | Yes | Bulk, Single Cell | Python | This work |
| CellChat | Mass-action-based probability | Runs separate analyses of each context, does pairwise comparisons and harmonizes them through a joint manifold learning | 2 | Yes | Single Cell | R | 10 |
| CellPhoneDB | Expression Mean | None | 1 | Yes | Single Cell | Python | 19 |
| CellTalker | Differential Combinations | Differential analysis between two contexts | 2 | No | Single Cell | R | 8 |
| Connectome | Modified Expression Product | Differential analysis between two contexts. An overall analysis of cell-type importance can be done for more contexts | 2 | No | Single Cell | R | 11 |
| ICELLNET | Expression Product | None | 1 | Yes | Bulk, Single Cell | R | 74 |
| iTalk | Differential Combinations | Differential analysis between two contexts | 2 | No | Single Cell | R | 75 |
| NATMI | Expression Product and Normalized Expression Product | Differential analysis between two contexts | 2 | No | Bulk, Single Cell | Python | 9 |
| NicheNet | Personalized-PageRank-based score | None | 1 | No | Bulk, Single Cell | R | 55 |
| scAgeCom | Geometric Mean | Differential analysis between two contexts | 2 | Yes | Single Cell | R | 76 |
| scTensor | Expression Product | None | 1 | No | Single Cell | R | 77 |
| SingleCellSignalR | Regularized Expression Product | None | 1 | No | Single Cell | R | 20 |

[a]For further details about distinct communication scores, see ref. 5 and/or respective references for each tool.
[b]Dependent on computational resources (e.g., memory availability).
LR, ligand-receptor.

of Tensor-cell2cell. Specifically, we analyzed twelve bronchoalveolar lavage fluid (BALF) samples from patients with different severities of COVID-19 (healthy, moderate and severe) with each method listed above. We assessed the consistency of decomposition between all five scoring methods by using the CorrIndex[21]. The CorrIndex value lies between 0 and 1, with a higher score indicating more dissimilar decomposition outputs; we thus report our similarity results as (1-CorrIndex). Our results indicate that Tensor-cell2cell can consistently identify context-dependent communication patterns independent of the initial communication scoring method (Fig. 3a, Supplementary Fig. 4), with a mean similarity score of 0.82. Furthermore, differences in decomposition results are driven at the ligand-receptor resolution, yet tend not to propagate to the cell- or context-resolution (Supplementary Notes and Supplementary Figs. 5 and 6). While these results agree with previous reports regarding the inconsistency of scoring methods for ligand-receptor interactions[22], they also show the power of tensor decomposition to resolve these inconsistencies

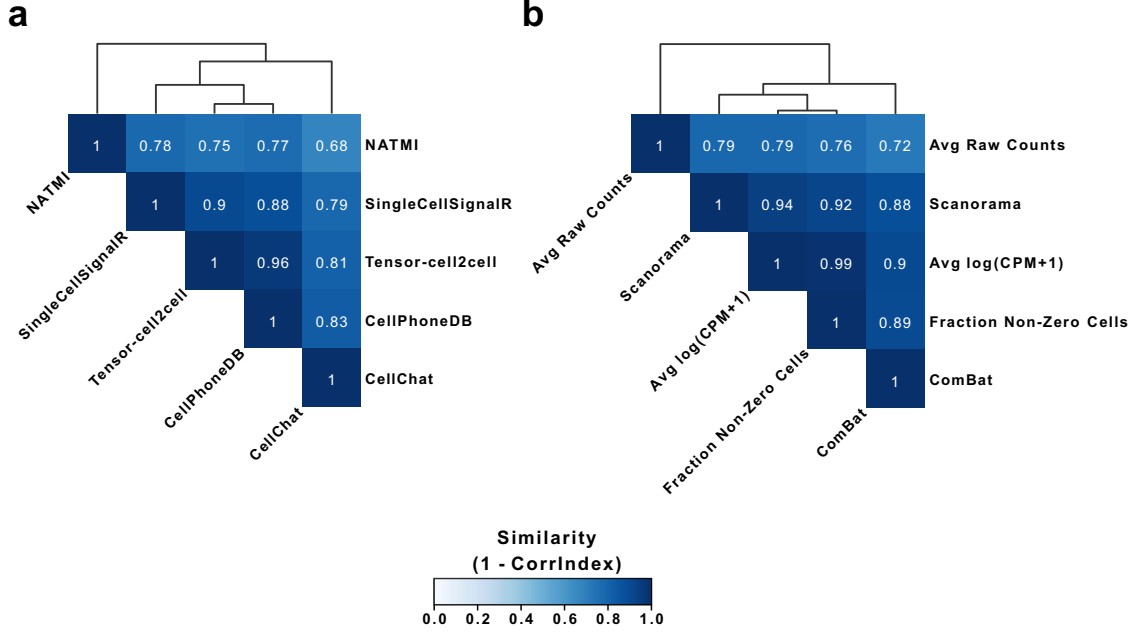

**Fig. 3 Comparison of tensor decompositions resulting from varying input values.** The similarity of tensor decompositions performed on 4D-communication tensors constructed from the single-cell dataset of BALF in patients with varying severities. For a given comparison, constructed tensors have the same elements in each dimension. **a** Similarity between tensor decompositions performed on 4D-communication tensors, each corresponding to communication scores computed from different tools for inferring cell–cell communication. The scoring functions correspond to those of CellChat[10], CellPhoneDB[19], NATMI[9], SingleCellSignalR[20] and the built-in methods in Tensor-cell2cell. **b** Similarity between tensor decompositions performed on 4D-communication tensors, each modifying the gene expression values by different preprocessing methods (log(CPM + 1) and the fraction of non-zero cells[23]) or batch-effect correction methods (Combat[24] and Scanorama[25]), as well as using the raw counts. The communication scores in (**b**) were calculated as the mean expression between the partners in each LR pair, previously aggregating gene expression at the single-cell level into the cell-type level. In (**a**, **b**) similarity was measured as (1-CorrIndex), where the CorrIndex[21] is a distance metric for comparing different decompositions on tensors containing the same indices and its values range from 0 to 1 (more similar to more dissimilar). Assessed methods were hierarchically clustered by the similarities of their tensor decompositions. Similarity values are provided in the Source Data file.

and identify biologically and conceptually consistent communication patterns.

Since Tensor-cell2cell requires the use of multiple conditions or samples, we also assessed biases that may have been introduced by batch effects during gene expression count transformation (e.g., normalization, batch correction, etc). Specifically, we assessed the impact of applying the log(CPM + 1) and the fraction of non-zero cells as preprocessing methods[23], and ComBat[24] and Scanorama[25] as batch-effect correction. Here, we also used the BALF COVID-19 samples and built the 4D-tensors using the gene expression values obtained in each case. After running the tensor decomposition, these strategies generated results that seem biologically comparable, as measured with a mean similarity score of 0.86 (Fig. 3b). As expected, using the raw counts leads to the most biased and different results in comparison to the other preprocessing methods; the mean similarity score between raw counts and all other approaches is 0.77. In contrast, the highest similarity was between the log(CPM + 1) and the non-zero fraction of cells. This result is also expected since the non-zero fraction of cells is comparable to the log(CPM + 1). However, the non-zero fraction performs better in comparisons of lowly expressed genes[23] (e.g., receptors on the cell surface[26]), so we included this fraction as part of the Tensor-cell2cell built-in workflow. Thus, Tensor-cell2cell can detect consistent CCC signatures independent of the method by which gene expression is corrected, with the exception of raw counts, as indicated by the high similarities observed (Fig. 3b).

**Tensor-cell2cell links intercellular communication with varying severities of COVID-19.** Great strides have been made to

unravel molecular and cellular mechanisms associated with SARS-CoV-2 infection and COVID-19 pathogenesis. Thus, we tested our method on a single-cell dataset of BALF samples from COVID-19 patients[27] to see how many cell–cell and LR pair relationships in COVID-19 could be revealed by Tensor-cell2cell. By decomposing the tensor associated with this dataset into 10 factors (Fig. 4a and Supplementary Fig. 1b), Tensor-cell2cell found factors representing communication patterns that are highly correlated with COVID-19 severity (Fig. 4c) and other factors that distinguish features of the different disease stages (Supplementary Fig. 7), consistent with the high performance that the classifier achieved for this dataset (Supplementary Fig. 3f,h). Furthermore, these factors involve signaling molecules previously linked with severity in separate works (Supplementary Table 3).

The first two factors capture CCC involving autocrine and paracrine interactions of epithelial cells with immune cells in BALF (Fig. 4a). The sample loadings of these factors reveal a communication pattern wherein the involved LR and cell–cell interactions become stronger as severity increases (Spearman correlation of 0.72 and 0.61, Fig. 4c and Supplementary Fig. 7). Although this observation was not reported in the original study, it is consistent with a previous observation of a correlation between COVID-19 severity and the airway epithelium-immune cell interactions[28]. Specifically, epithelial cells are highlighted by Tensor-cell2cell as the main sender cells in factor 1 (Fig. 4a), and we further provide details of the molecular mechanisms involving top-ranked signals such as APP, MDK, MIF and CD99 (Fig. 4b). These molecules have been reported to be produced by epithelial cells[29–35] and participate in immune cell recruiting[31–33,36], in response to mechanical stress in lungs[34] and regeneration of the

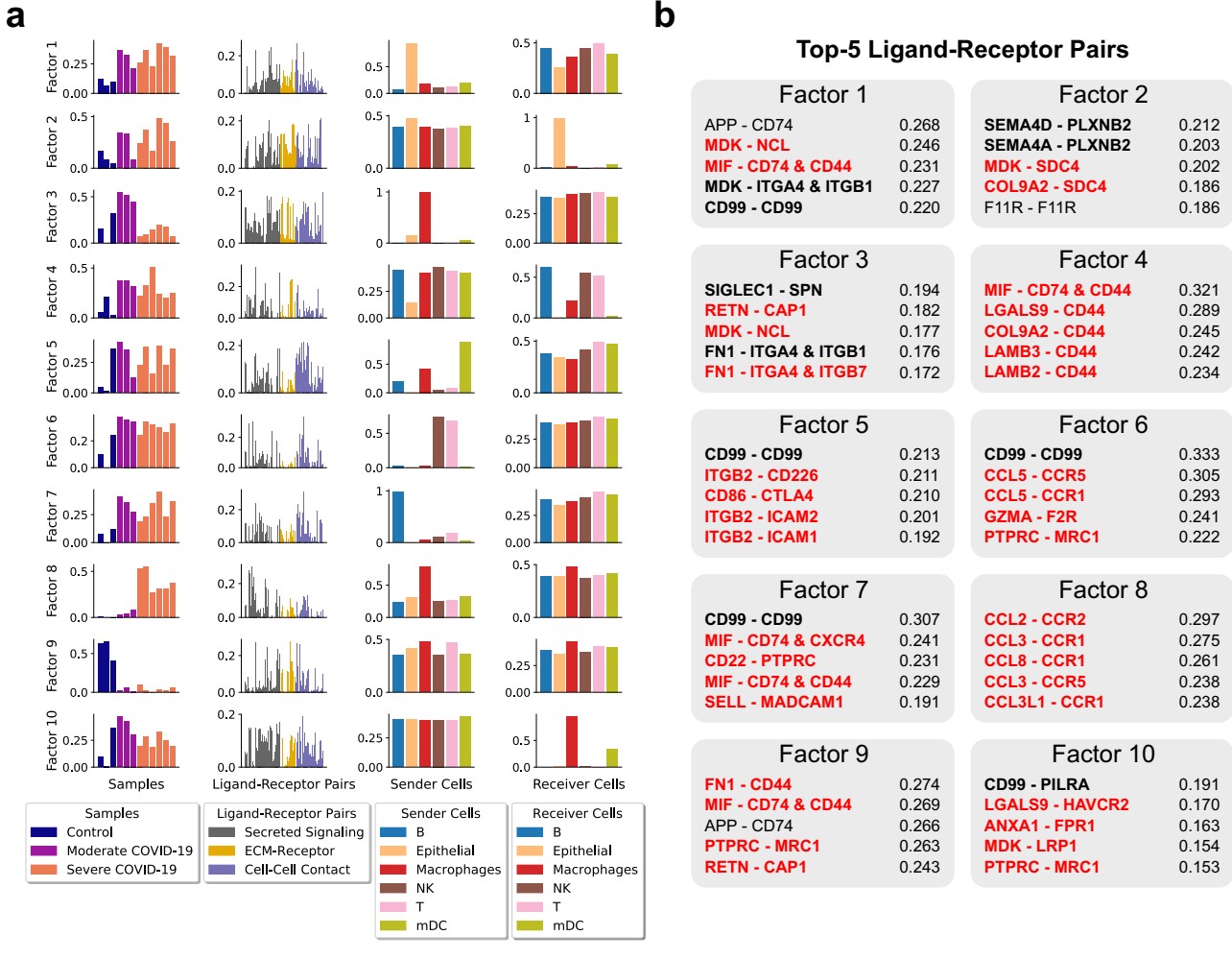

alveolar barrier during viral infection[35]. In addition, epithelial cells act as the main receiver in factor 2 (Fig. 4a), involving proteins such as PLXNB2, SDC4 and F11R (Fig. 4b), which were previously determined important for tissue repair and inflammation during lung injury[37–39]. Remarkably, a new technology for experimentally tracing CCC revealed that SEMA4D-PLXNB2 interaction promotes inflammation in a diseased central nervous system[40]; our approach suggests a similar role promoting

inflammation in severe COVID-19, specifically mediating the communication between immune and epithelial cells, as reflected in factor 2 (Fig. 4b).

Our strategy also elucidates communication patterns attributable to specific groups of patients according to disease severity (Fig. 4a). For example, we found interactions that are characteristic of severe (factor 8) and moderate COVID-19 (factors 3 and 10), and healthy patients (factor 9) (adj. P-value < 0.05,

**Fig. 4 Deconvolution of intercellular communication in patients with varying severity of COVID-19. a** Factors obtained after decomposing the 4D-communication tensor from a single-cell dataset of BALF in patients with varying severities of COVID-19. 10 factors were selected for the analysis, as indicated in Supplementary Fig. 1b. Here, the context corresponds to samples coming from distinct patients (12 in total, with three healthy controls, three moderate infections, and six severe COVID-19 cases). Each row represents a factor and each column represents the loadings for the given tensor dimension (samples, LR pairs, sender cells and receiver cells), normalized to unit Euclidean length. Bars are colored by categories assigned to each element in each tensor dimension, as indicated in the legend. **b** List of the top 5 ligand-receptor pairs ranked by loading for each factor. The corresponding ligands and receptors in these top-ranked pairs are mainly produced by sender and receiver cells with high loadings, respectively. Ligand-receptor pairs with supporting evidence (Supplementary Table 3) for a relevant role in general immune response (black bold) or in COVID-19-associated immune response (red bold) are highlighted. **c** Coefficients associated with loadings of each factor: Spearman coefficient quantifying correlation between sample loadings and COVID-19 severity, and Gini coefficient quantifying the dispersion of the edge weights in each factor-specific cell–cell communication network (to measure the imbalance of communication). Important values are highlighted in red (higher absolute Spearman coefficients represent stronger correlations; while smaller Gini coefficients represent distributions with similar edge weights). Loadings and coefficients are provided in the Source Data file.

Supplementary Fig. 7). Factor 8 was the most correlated with severity of the disease (Spearman coefficient 0.92, Fig. 4c) and highlights macrophages playing a major role as pro-inflammatory sender cells. Their main signals include CCL2, CCL3 and CCL8, which are received by cells expressing the receptors CCR1, CCR2 and CCR5 (Fig. 4b). Consistent with our result, another study of BALF samples[28] revealed that critical COVID-19 cases involve stronger interactions of cells in the respiratory tract through ligands such as CCL2 and CCL3, expressed by inflammatory macrophages[28]. Moreover, the inhibition of CCR1 and/or CCR5 (receptors of CCL2 and CCL3) has been proposed as a potential therapeutic target for treating COVID-19[28,41]. Tensor-cell2cell also deconvolved patterns attributable to moderate rather than severe COVID-19, also highlighting interactions driven by macrophages (factors 3 and 10; Fig. 4a). However, top-ranked molecules (Fig. 4b) and gene expression patterns (Supplementary Fig. 8) suggest that the intercellular communication is led by macrophages with an anti-inflammatory M2-like phenotype, in contrast to factor 8 (pro-inflammatory phenotype). Multiple top-ranked signals in factors 3 and 10 have been associated with an M2 macrophage phenotype acting in the immune response to SARS-CoV-2[42–47].

In contrast to severe and moderate COVID-19 patients, communication patterns associated with healthy subjects involve all sender-receiver cell pairs with a similar importance. In particular, factor 9 (Fig. 4a) demonstrated the smallest Gini coefficient (0.09; Fig. 4c), which measures the extent to which edge weights between sender and receiver cells are evenly distributed in the factor-specific cell–cell communication network. Smaller Gini coefficients show more even distributions, i.e., more equally weighted potential of communication across sender and receiver cell pairs (see Methods). This indicates that the intercellular communication represented by factor 9 is ubiquitous across cell types. Thus, this conservation across cells may be an indicator of communication during homeostasis, since the context loadings for this factor are not associated with disease (Supplementary Fig. 7). Interestingly, a top-ranked LR pair in factor 9 is MIF-CD74/CD44 (Fig. 4b), which is consistent with ubiquitous expression of MIF across tissues and its protective role in normal conditions[35,48]. Thus, Tensor-cell2cell extracts communication patterns distinguishing one group of patients from another and detects known mechanisms of immune response during disease progression (Supplementary Notes), which is important for therapeutic applications.

**Tensor-cell2cell elucidates communication mechanisms associated with Autism Spectrum Disorders**. Dysregulation of neurodevelopment in Autism Spectrum Disorders (ASD) is associated with perturbed signaling pathways and CCC in complex ways[49]. To understand these cellular and molecular mechanisms, we analyzed single-nucleus RNA-seq (snRNA-seq)

data from postmortem prefrontal brain cortex (PFC) from 13 ASD patients and 10 controls[50]. We built a 4D-communication tensor containing 16 cell types present in all samples, including neurons and non-neuronal cells, and 749 LR pairs; then we used Tensor-cell2cell to deconvolve their associated CCC into 6 context-driven patterns (Fig. 5a and Supplementary Fig. 1c). In these factors, we observe communication between all neurons (factor 1), as well as communication of specific neurons in the cortical layers I–VI (factors 2 and 3), interneurons (factor 4), astrocytes and oligodendrocytes (factor 5), and endothelial cells (factor 6).

Tensor-cell2cell's outputs can be further dissected using downstream analyses with common approaches. To illustrate this, we ranked the LR pairs by their loadings in a factor-specific fashion, and ran Gene Set Enrichment Analysis[51] (GSEA) using LR pathway sets built from KEGG pathways[52] (see Methods). We observed that each factor was associated with different biological functions including axon guidance, cell adhesion, extracellular-matrix-receptor interaction, ERBB signaling, MAPK signaling, among others (Fig. 5b). Dysregulation of axon guidance, synaptic processes and MAPK pathway have been previously linked to ASD from differential analysis[50,53], supporting our observations. Moreover, our results extend to other roles associated with extracellular matrix, focal adhesion of cells, regulation of actin cytoskeleton, and signaling through ErbB receptors, which involves Akt, PI3K, and mTOR pathways, as well as regulation of cell proliferation, migration, motility, differentiation, and apoptosis[54]. Thus, Tensor-cell2cell outputs can be used to assign macro-scale biological functions to each of the factors, extending the interpretation of factor-specific CCC.

After identifying main pathways involved in each factor, one can further use sample loadings to identify how these functions are associated with each sample group. By doing so, we found that factors 3 and 4 significantly distinguish ASD from typically developing controls (Fig. 5c). Neurons in cortical layers are the main sender cells in factor 3, while interneurons are key receiver cell types in factor 4 (Fig. 5a and Supplementary Fig. 9), with parvalbumin interneurons (IN-PV), and SV2C-expressing interneurons (IN-SV2C) as the top-ranked cells, consistent with the previously reported cell types that are more affected in ASD condition[50] (i.e., with a greater number of dysregulated genes), and that correspond to neurons in the cortical layers I–VI, IN-SV2C and IN-PV. Thus, considering the overall decreased sample loadings in the ASD group, the GSEA results, and the factor-specific CCC networks built from the cell loadings (Supplementary Fig. 9), our analysis suggests that there is a downregulation of axon guidance, cell adhesion, and ERBB signaling involving neurons in cortical layers I–VI and interneurons in ASD patients. See Supplementary Notes for further discussion.

Clustering methods can be applied for grouping samples in an unsupervised manner. Thus, we can assess the overall similarity

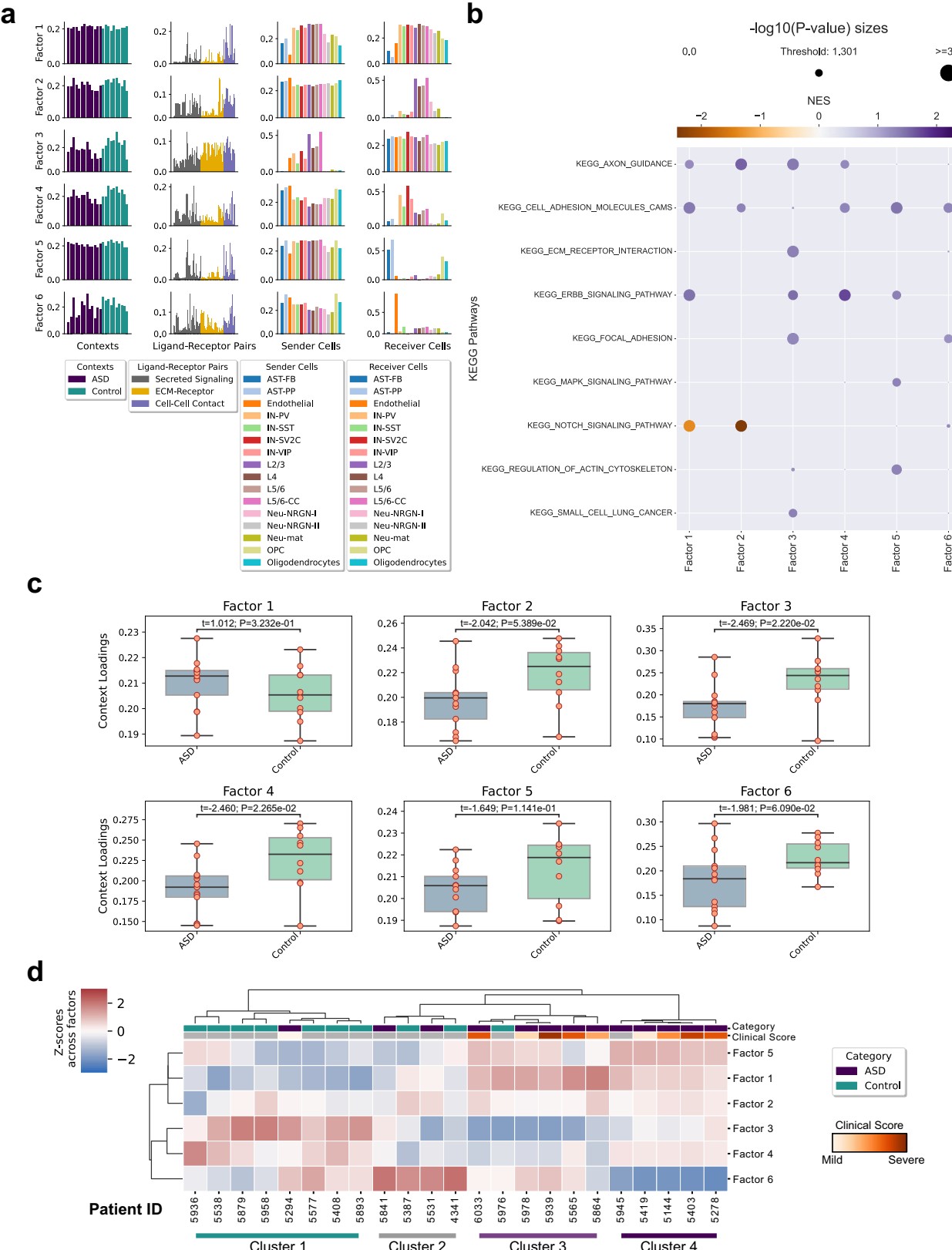

between samples across all factors; considering combinations of factors can offer additional insights to the analysis as compared to considering one factor at a time. We use hierarchical clustering to group samples into four main clusters (Fig. 5d). Cluster 1 mainly groups controls, cluster 2 is not associated with any category, cluster 3 mostly represents ASD patients, and cluster 4 is

completely related to ASD condition. These clusters also reveal that combinations of factors separate samples by ASD and control groups. For example, samples in cluster 1 seem to have smaller loadings in factors 1 and 5, and higher loadings in factors 3 and 4. Interestingly, the only ASD sample present in this cluster had the smallest ASD clinical score, suggesting that CCC

**Fig. 5 Application of Tensor-cell2cell to study mechanisms underlying intercellular communication in patients with ASD. a** Factors obtained after decomposing the 4D-communication tensor from a single-nucleus dataset of prefrontal brain cortex samples from patients with or without ASD. Six factors were selected for the analysis, as indicated in Supplementary Fig. 1c. Here, the context corresponds to samples coming from distinct patients ($n = 23$, thirteen ASD patients and ten controls). Each row represents a factor and each column represents the loadings for the given tensor dimension (samples, LR pairs, sender cells and receiver cells), normalized to unit Euclidean length. Bars are colored by categories assigned to each element in each tensor dimension, as indicated in the legend. Cell-type annotations are those used in ref. [50]. **b** GSEA performed on the pre-ranked LR pairs by their respective loadings in each factor, and using KEGG pathways. Dot sizes are proportional to the negative logarithmic of the $P$-values, as indicated at the top of the panel. The threshold value indicates the size of a $P$-value = 0.05. The dot colors represent the normalized enrichment score (NES) after the permutations performed by the GSEA, as indicated by the colorbar. $P$-values were obtained from the permutation step performed by GSEA, and adjusted with a Benjamini–Hochberg correction across all factors. **c** Boxplot representation for ASD ($n = 13$) and control ($n = 10$) groups of patients. Each panel represents the sample loadings, grouped by condition category, in each of the factors. Boxes represent the quartiles and whiskers show the rest of each distribution. Groups were compared by a two-sided independent $t$-test, followed by a Bonferroni correction. For each pairwise comparison, the exact values of the test statistics ($t$) and the adjusted $P$-values ($P$) are shown. **d** Heatmap of the standardized sample loadings across factors ($z$-scores) for each of the samples. Samples and factors were grouped by hierarchical clustering. Major clusters of the samples are indicated at the bottom. The category of each sample is colored on the top, according to the legend. A clinical score of each patient is also shown, according to the colorbar. Controls, and ASD samples without an assigned score, were colored gray. This clinical score summarizes the social interactions, communication, repetitive behaviors, and abnormal development of the patients, as indicated in ref. [50]. Loadings, enrichment scores, and clinical scores are provided in the Source Data file.

mechanisms are more similar to controls when the phenotype is mild. In contrast, cluster 3 shows an opposite CCC behavior to cluster 1. Cluster 4 also reveals that the combination of factor 6 with low sample loadings and factors 1 and 5 with high values is a strong marker of several ASD patients, even though factors 1, 5, and 6 did not show significant differences between sample groups (Fig. 5c). Based on this, patients in cluster 4 had increased CCC through the NRXNs-NRLGs, CTNs-NRCAMs, and NCAMs-NCAMs interactions (synapse and cell adhesion) in neurons as senders and receivers, and astrocytes and oligodendrocytes as receivers, as well as a decreased CCC through VEGFs-FLT1, PTPRM-PTPRM, and PTN-NCL interactions (angiogenesis, neural migration and neuroprotection) related to endothelial cells as the main receivers (Supplementary Table 4). Finally, both ASD-clusters seem to be slightly distinct in terms of phenotype, considering their mean clinical scores of 25.0 and 22.8, respectively for clusters 3 and 4, but without significant differences. Thus, downstream analyses reveal that multiple dysregulations of CCC patterns captured by Tensor-cell2cell occur simultaneously in ASD condition (Fig. 5d), even though these patterns could not explain phenotypic differences when considered in isolation (Fig. 5c).

## Discussion
Here we present Tensor-cell2cell, a computational approach that identifies modules of cell–cell communication and their changes across contexts (e.g., across subjects with different disease severity, multiple time points, different tissues, etc.). Our approach can rank LR pairs based on their contribution to each communication module and connect these signals to specific cell types and phenotypes. Tensor-cell2cell's ability to consider multiple contexts simultaneously to identify context-dependent communication patterns goes beyond state-of-the-art tools, which are either unaware of the context driving CCC[5,19,55,56] or require analysis of each context separately to perform pairwise comparisons in posterior steps[10,11]. Tensor-cell2cell is therefore a flexible method that can integrate multiple datasets and readily identify patterns of intercellular communication in a context-aware manner, reporting them through interconnected and easily interpretable scores.

Tensor-cell2cell robustly detects communication patterns using many other scoring methods[13]. Thus, our method is not only an improvement over other tools, but also greatly extends these tools, enabling unique analyses with existing methods. One can choose any tool of interest, run it on each context separately, and use the resulting communication scores to build and deconvolve a

4D-communication tensor. Other tools, such as CellChat, allow the generation of scores at the signaling pathway level instead of LR pairs. This, combined with Tensor-cell2cell, could provide additional information about changes in signaling pathways. Thus, Tensor-cell2cell can also be used for analyzing any other score linking gene expression from cell pairs, beyond just scores based on protein-protein interactions. In this regard, our tool outputs consistent results regardless of the preprocessing and batch correction method we evaluated (Fig. 3b). Nevertheless, it is best practice to employ integration/batch-correction methods to correct gene expression and annotate cell types before running Tensor-cell2cell to ensure this source of variation is controlled[57].

Tensor-cell2cell is faster for analyzing multiple samples than pairwise comparisons, providing a considerable improvement in running time and reduced memory requirements (Supplementary Notes). Tensor-cell2cell's runtime can be further accelerated when a GPU is available (Supplementary Fig. 3a). It is also more accurate, resulting in 10–20% higher classification accuracy of subjects with COVID-19 when compared to CellChat (Supplementary Fig. 3e–h). However, we note that benchmarking CCC prediction tools is challenging due to the lack of a ground truth[5], and it is hard to compare and evaluate tools because of the qualitative differences in their outputs[22] (Supplementary Notes). While pairwise comparisons can be informative about differential cellular and molecular mediators of communication, the results are less interpretable (Supplementary Figs. 10–13), do not provide the multi-scale resolution available in Tensor-cell2cell (Figs. 4a and 5a), and do not identify context-dependent patterns.

Meaningful biology can be easily identified from Tensor-cell2cell. For example, a manual interpretation of the BALF COVID-19 decomposition found communication results not previously observed in the original study[27] and recapitulated findings spanning tens of peer-reviewed articles (Supplementary Table 3). This included a correlation between the lung epithelium-immune cell interactions and COVID-19 severity[28] and molecular mediators that distinguished moderate and severe COVID-19 (see "Tensor-cell2cell elucidates molecular mechanisms distinguishing moderate from severe COVID-19" in the Supplementary Notes). Additionally, Tensor-cell2cell results can be coupled with downstream analysis methods to facilitate interpretation and provide further insights of underlying mechanisms. In our ASD case-study (Fig. 5), such analyses included GSEA, clustering, visualization and statistical comparison of factors, and factor-specific analysis of sender-receiver communication networks (Supplementary Fig. 9). In the ASD case-study, we found dysregulated CCC directly distinguished

ASD patients from controls and was linked with a down-regulation of axon guidance, cell adhesion, synaptic processes, and ERBB signaling in cortical neurons and interneurons (Fig. 5a, b), consistent with previous evidence[50,53,58,59]. Moreover, accounting for the combinatorial relationship of samples across factors demonstrated additional complex relationships of CCC dysregulation (Fig. 5d).

A limitation to consider is the potential of missing communication scores in the tensor (e.g., when a rare cell type appears in only one sample). Although Tensor-cell2cell can handle cell types that are missing in some conditions, the implemented tensor decomposition algorithm can be further optimized for missing values. Since the implemented algorithm is not optimized for this purpose, we built a 4D-communication tensor that contains only the cell types that are shared across all samples in our COVID-19 and ASD study cases. Thus, further developments will facilitate analyses with missing values to include all possible members of communication (i.e., LR pairs and cell types that may be missing in certain contexts).

In addition to single-cell data analyzed here, Tensor-cell2cell also accepts bulk transcriptomics data (an example of a time series bulk dataset of *C. elegans* is included in a Code Ocean capsule, see Methods), and it could further be used to analyze proteomic data. We demonstrated the application of Tensor-cell2cell in cases where samples correspond to distinct patients, but it can be applied to many other contexts. For instance, our strategy can be readily applied to time series data by considering time points as the contexts, and to spatial transcriptomic datasets, by previously defining cellular niches or neighborhoods as the contexts, given their spatial signatures[60]. We have included Tensor-cell2cell as a part of our previously developed tool cell2cell[61], enabling previous functionalities such as employing any list of LR pairs (including protein complexes), multiple visualization options, and personalizing the communication scores to account for other signaling effects such as the (in)activation of downstream genes in a signaling pathway[55,62,63]. Thus, these attributes make Tensor-cell2cell valuable for identifying key cell–cell and LR pairs mediating complex patterns of cellular communication within a single analysis for a wide range of studies.

## Methods

**RNA-seq data processing**. RNA-seq datasets were obtained from publicly available resources. Datasets correspond to a large-scale single-cell atlas of COVID-19 in humans[64], a COVID-19 dataset of single-cell transcriptomes for BALF samples[27]. COVID-19 datasets were collected as raw count matrices from the NCBI's Gene Expression Omnibus[65] (GEO accession numbers "GSE158055" and "GSE145926", respectively), while the ASD dataset is available in the NCBI's BioProject under accession code "PRJNA434002", but we obtained the log2-transformed UMI counts from the "project website [https://cells.ucsc.edu/autism/downloads.html]". In total, the first dataset contains 1,462,702 single cells, the second 65,813 and the last one 104,559 single nuclei. The first dataset contains samples of patients with varying severities of COVID-19 (control, mild/moderate and severe/critical) and we selected just 60 PBMC samples among all different sample sources (20 per severity type). In the second dataset, we considered the 12 BALF samples of patients with varying severities of COVID-19 (3 control, 3 moderate and 6 severe) and preprocessed them by removing genes expressed in fewer than 3 cells, which left a total of 11,688 genes in common across samples. In the ASD dataset, PFC samples from 23 patients with and without ASD condition (13 ASD patients and 10 controls) were considered, and preprocessed similarly to the BALF dataset, resulting in a total of 24,298 genes in common across samples. In all datasets, we used the cell type labels included in their respective metadata. We aggregated the gene expression from single cells/nuclei into cell types by calculating the fraction of cells in the respective label with non-zero counts, as previously recommended for properly representing genes with low expression levels[23], as usually happens with genes encoding surface proteins[26].

**Ligand-receptor pairs**. A human list of 2,005 ligand-receptor pairs, 48% of which include heteromeric-protein complexes, was obtained from CellChat[10]. We filtered this list by considering the genes expressed in the PBMC and BALF expression datasets and that match the IDs in the list of LR pairs, resulting in a final list of

1639 and 189 LR pairs, respectively. While in the ASD dataset, 749 LR pairs that matched the gene IDs were considered.

**Building the context-aware communication tensor**. For building a context-aware communication tensor, three main steps are followed: (1) A communication matrix is built for each ligand-receptor pair contained in the interaction list from the gene expression matrix of a given sample. To build this communication matrix, a communication score[5] is assigned to a given LR pair for each pair of sender-receiver cells. The communication score is based on the expression of the ligand and the receptor in the respective sender and receiver cells (Fig. 1a). (2) After computing the communication matrices for all LR pairs, they are joined into a 3D-communication tensor for the given sample (Fig. 1b). Steps 1 and 2 are repeated for all the samples (or contexts) in the dataset. (3) Finally, the 3D-communication tensors for each sample are combined, each of them representing a coordinate in the 4th-dimension of the 4D-communication tensor (or context-aware communication tensor; Fig. 1c).

To build the tensor for all datasets, we computed the communication scores as the mean expression between the ligand in a sender cell type and cognate receptor in a receiver cell type, as previously described[19]. For the LR pairs wherein either the ligand or the receptor is a multimeric protein, we used the minimum value of expression among all subunits of the respective protein to compute the communication score. In all cases we further considered cell types that were present across all samples. Thus, the 4D-communication tensor for the PBMC, BALF and ASD datasets resulted in a size of $60 \times 1639 \times 6 \times 6$; $12 \times 189 \times 6 \times 6$, and $23 \times 749 \times 16 \times 16$ respectively (that is, samples x ligand-receptor pairs x sender cell types x receiver cell types).

**Non-negative tensor component analysis**. Briefly, non-negative TCA is a generalization of NMF to higher-order tensors (matrices are tensors of order two). To detail this approach, let $\chi$ represent a $C \times P \times S \times T$ tensor, where $C$, $P$, $S$ and $T$ correspond to the number of contexts/samples, ligand-receptor pairs, sender cells and receiver cells contained in the tensor, respectively. Similarly, let $\chi_{ijkl}$ denote the representative interactions of context $i$, using the LR pair $j$, between the sender cell $k$ and receiver cell $l$. Thus, the TCA method underlying Tensor-cell2cell corresponds to CANDECOMP/PARAFAC[66,67], which yields the decomposition, factorization or approximation of $\chi$ through a sum of $R$ tensors of rank-1 (Fig. 1d):

$$\chi \approx \sum_{r=1}^{R} c^r \otimes p^r \otimes s^r \otimes t^r \qquad (1)$$

Where the notation $\otimes$ represents the outer product and $c^r$, $p^r$, $s^r$ and $t^r$ are vectors of the factor $r$ that contain the loadings of the respective elements in each dimension of the tensor (Fig. 1e). These vectors have values greater than or equal to zero. Similar to NMF, the factors are permutable and the elements with greater loadings represent an important component of a biological pattern captured by the corresponding factor. Values of individual elements in this approximation are represented by:

$$\chi_{ijkl} \approx \sum_{r=1}^{R} c_i^r \otimes p_j^r \otimes s_k^r \otimes t_l^r \qquad (2)$$

The tensor factorization is performed by iterating the following objective function until convergence through an alternating least squares minimization[17,68]:

$$\min_{\{c,p,s,t\}} \left\| \chi - \sum_{r=1}^{R} c^r \otimes p^r \otimes s^r \otimes t^r \right\|_F^2 \qquad (3)$$

Where $\|\bullet\|_F^2$ represent the squared Frobenius norm of a tensor, calculated as the sum of element-wise squares in the tensor:

$$\|\chi\|_F^2 = \sum_{i=1}^{C} \sum_{j=1}^{P} \sum_{k=1}^{S} \sum_{l=1}^{T} \chi_{ijkl}^2 \qquad (4)$$

All the described calculations were implemented in Tensor-cell2cell through functions available in Tensorly[69], a Python library for tensors.

**Measuring the error of the tensor decomposition**. Depending on the number of factors used for approximating the 4D-communication tensor, the reconstruction error calculated in the objective function can vary. To quantify the error with an interpretable value, we used a normalized reconstruction error as previously described[12]. This normalized error is on a scale of zero to one and is analogous to the fraction of unexplained variance used in PCA:

$$\frac{\left\| \chi - \sum_{r=1}^{R} c^r \otimes p^r \otimes s^r \otimes t^r \right\|_F^2}{\|\chi\|_F^2} \qquad (5)$$

**Running tensor-cell2cell with communication scores from external tools**. We assessed the similarity of tensor decomposition on the BALF dataset using different communication scoring methods (CellChat[10], CellPhoneDB[19], NATMI[9], SingleCellSignalR[20], and Tensor-cell2cell's built-in scoring). To enable consistency between methods, we used the same ligand-receptor PPI database (CellChat—see

"Ligand-receptor pairs") and ran each method via LIANA[22]. LIANA provides a number of advantages over running each tool separately, including consistent thresholding and parameters, interoperability between methods and LR databases, and modifications to allow methods that could not originally account for protein complexes to do so. We adjusted parameters to match those of Tensor-cell2cell's built-in scoring by not filtering for minimal proportions of expression by cell type or thresholding for differentially expressed genes.

As input to LIANA, we constructed a Seurat object with log(CPM + 1) normalized counts for each sample. For each tool and sample, LIANA outputs an edge-list of communication scores for a given combination of sender and receiver cells, as well as ligand-receptor pairs. We extended Tensor-cell2cell's functionalities to restructure a set of these edge-lists, each associated with a sample, into a 4D-communication tensor (Fig. 1). This functionality enables users to either provide input expression matrices and use Tensor-cell2cell's built-in scoring, or to run their communication scoring method of choice on each sample and provide the resultant edge-lists as input. To further ensure consistency, we subset each resultant tensor to the intersection of ligand-receptor pairs scored across all 5 methods. For each method, this resulted in a tensor consisting of 12 samples, 172 ligand-receptor pairs, and 6 sender and receiver cells.

### Evaluating the effect of gene expression preprocessing and batch-effect correction on Tensor-cell2cell.
To evaluate how gene expression preprocessing and batch-effect correction impact the results of Tensor-cell2cell, we assessed the similarity of tensor decomposition on the BALF dataset. To compute the communication scores for building the tensors (Fig. 1a), we used different gene expression values, including the raw UMI counts, the preprocessed values with log(CPM + 1) and the fraction of non-zero cells[23], and the batch-corrected values with ComBat[24] and Scanorama[25]. Except by the fraction of non-zero cells, which already aggregated single-cells into cell-types, other values were aggregated into the cell-type level by computing the average value for each gene across single cells with the same cell-type label. As the communication score, we used the expression mean of the interacting partners in each LR pair. Thus, we built 4D-communication tensors as mentioned for the BALF data in the Methods subsection "Building the context-aware communication tensor". The tensor decomposition resulting with the fraction of non-zero cells in this case corresponds to the same in Fig. 4.

### Measuring the similarity between distinct tensor decomposition runs.
To assess decomposition consistency between different scoring methods or pre-processing pipelines, we employed the CorrIndex[21]. The CorrIndex is a permutation- and scaling-invariant distance metric that enables consistent comparison of decompositions between tensors containing the same elements, without need to align the factors obtained in each case (separate tensor decompositions can output similar factors but in different order). The CorrIndex value lies between 0 and 1, with a higher score indicating more dissimilar decomposition outputs. To score tensor decompositions, the output factor matrices must first be vertically stacked. We implemented a modification that instead assesses each tensor dimension separately (see Supplementary Note for more details). While taking the minimal score between all dimensions tends to be more stringent, it disregards the combinatorial effects of all dimensions together. These combinatorial effects are important because they better reflect the goal of tensor decomposition and because similarity in those dimensions that are not the minimal one may be artificially inflated. To facilitate the use of the CorrIndex and its modified version, we wrote a Python implementation that is available on the Tensorly package[69].

### Downstream analyses using the loadings from the tensor decomposition.
We incorporate several downstream analyses of Tensor-cell2cell's decomposition outputs to further elucidate the underlying cell- and molecular- mediators of cell–cell communication. Each of these analyses are associated with a specific tensor dimension, and thus, a specific biological resolution. This includes (1) statistical, correlative, and clustering analyses to understand context associations for each factor, (2) gene set enrichment analysis of ligand-receptor loadings to identify granular signaling pathways associated with factors, (3) the generation of factor-specific cell–cell communication networks to represent the overall communication state of cells in that factor.

We can understand the context associations for a factor by comparing the loadings of samples associated with distinct contexts. For statistical significance, we conduct an independent t-test pairwise between each context group associated with the samples and use Bonferonni's correction to account for multiple comparisons. We use this for both the COVID-19 BALF dataset (Supplementary Figures 7 and 8) and the ASD dataset (Fig. 5c). We also conduct correlative analyses – assuming ordinal contexts (i.e., healthy control < moderate COVID-19 < severe COVID-19), we take the Spearman correlation between the sample loadings and sample severity (Fig. 4c). Finally, we also hierarchically cluster the samples using their loadings across all factors (Fig. 5d). For this purpose, we use the normalized loadings resulting from the tensor decomposition, and standardize them across all factors. Then, we apply an agglomerative hierarchical clustering by using Ward's method and the Euclidean distance as a metric. Note that this type of clustering analysis can be applied to the other tensor dimensions.

We can use the LR-pair loadings of a factor to identify the signaling pathways associated with it, by using the Gene Set Enrichment Analysis[51] (GSEA). Before running the analysis, pathways of interest have to be assigned to a list of associated LR pairs. We do that by considering the KEGG gene sets available at the "MsigDB[51] [http://www.gsea-msigdb.org/]". We annotate a LR pair available in CellChat with the gene sets that contain all genes participating in that LR interaction. Then, by filtering LR pathway sets to those containing at least 15 LR pairs, we end up with 22 LR pathway sets. To run GSEA, we rank the LR pairs in each factor by their loadings, and use the PreRanked GSEA function in the package gseapy, by including the 22 LR pathway sets as input. As parameters of the "gseapy.prerank" function, we consider 999 permutations, gene sets (LR pathway sets here) with at least 15 elements, and a score weight of 1 for computing the enrichment scores[51].

Finally, we generate factor-specific cell–cell communication networks. To do so, for a factor r, we take the outer product between the sender-cell loadings vector, $s^r$, and the receiver-cell loadings vector, $t^r$. Conceptually, this outer product represents an adjacency matrix of a factor-specific cell–cell communication network, where each value is an edge weight representing the overall communication between a pair of sender-receiver cells (Supplementary Fig. 9). We can further use this network to understand the communication distribution inequality between sender and receiver cells. We compute a Gini coefficient[70] ranging between 0 and 1 on the distribution of edge weights in the adjacency matrix (Fig. 4c). A value of 1 represents maximal inequality of overall communication between cell pairs (i.e. one cell pair has a high overall communication value while the others have a value of 0) and 0 indicates minimal inequality (i.e. all cell pairs have the same overall communication values). More generally, the outer product between any two tensor dimension loadings for a given factor conceptually represents the joint distribution of the elements in those two dimensions and can be informative of how the specific elements are related.

### Benchmarking of computational efficiency of tools.
We measured the running time and memory demanded by Tensor-cell2cell and CellChat to analyze the COVID-19 dataset containing PBMC samples. Each tool was evaluated in two scenarios: either using each sample individually, or by first combining samples by severity (control, mild/moderate, and severe/critical) by aggregating the expression matrices. The latter was intended to favor CellChat by diminishing the number of pairwise comparisons to always be between three contexts; thus, increases in running time or memory demand in this case are not due to an exponentiation of comparisons (n samples choose 2). CellChat was run by following the procedures outlined in the Comparison_analysis_of_multiple_datasets vignette in its "tutorial [https://github.com/sqjin/CellChat/tree/master/tutorial]". Briefly, signaling pathway communication probabilities were first individually calculated for each sample or context. Next, pairwise comparisons between each sample or context were obtained by computing either a "functional" or a "structural" similarity. The functional approach computes a Jaccard index to compare the signaling pathways that are active in two cellular communication networks, while the structural approach computes a network dissimilarity[71] to compare the topology of two signaling networks (see ref. [10] for further details). Finally, CellChat performs a manifold learning approach on sample similarities and returns UMAP embeddings for each signaling pathway in each different context (e.g., if CellChat evaluates 10 signaling pathways in 3 different contexts, it will return embeddings for 30 points) which can be used to rank the similarity of shared signaling pathways between contexts in a pairwise manner.

The analyses of computational efficiency were run on a compute cluster of 2.8 GHz ×2 Intel(R) Xeon(R) Gold 6242 CPUs with 1.5 TB of RAM (Micron 72ASS8G72LZ-2G6D2) across 32 cores. Each timing task was limited to 128 GB of RAM on one isolated core and one thread independently where no other processes were being performed. To limit channel delay, data was stored on the node where the job was performed, where the within socket latency and bandwidth were 78.9 ns and 46,102 MB/s respectively. For all timing jobs, the same ligand-receptor pairs and cell types were used. Furthermore, to make the timing comparable, all samples in the dataset were subsampled to have 2,000 single cells. In the case of Tensor-cell2cell, the analysis was also repeated by using a GPU, which corresponded to a Nvidia Tesla V100.

### Training and evaluation of a classification model.
A Random Forest[72] (RF) model was trained to predict disease status based on both COVID-19 status (healthy control vs. patient with COVID-19) and severity (healthy control, moderate symptoms, and severe symptoms). The RF model was trained using a Stratified K-Folds cross-validation (CV) with 3-Fold CV splits. On each CV split a RF model with 500 estimators was trained and RF probability-predictions were compared to the test set using the Receiver Operating Characteristic (ROC). The mean and standard deviation from the mean were calculated for the area under the Area Under the Curve (AUC) across the CV splits. This classification was performed on the context loadings of Tensor-cell2cell, and the two UMAP dimensions of the structural and functional joint manifold learning of CellChat, for both the BALF and PBMC COVID-19 datasets. All classification was performed through Scikit-learn (v. 0.23.2)[73].

**Statistics and reproducibility**. No sample-size calculation was performed. Instead, we used the number of samples included in each of the previously published datasets that we used. The only data exclusion performed was for the PBMC COVID-19 datasets, which originally includes 284 samples. For running our benchmarking, we subset the dataset to only include 60 samples. These samples were randomly selected for each COVID-19 severity, with 20 corresponding to control patients, 20 to mild/moderate COVID-19 patients, and 20 to severe/critical COVID-19 patients. For reproducibility, we deposited all our analyses including data and exact versions of code and software in a Code Ocean capsule. Results can be exactly replicated by running the analyses in that capsule. Randomization and blinding do not apply to this work because we analyzed previously published and annotated datasets.

**Reporting summary**. Further information on research design is available in the Nature Research Reporting Summary linked to this article.

## Data availability

All input data used for the analyses in this work and the result-generated data are available online in a "Code Ocean capsule [https://doi.org/10.24433/CO.0051950.v2]". In particular, we used a single-cell atlas of COVID-19 in humans[64], previously deposited in the NCBI's Gene Expression Omnibus database under accession code "GSE158055", a COVID-19 dataset of single-cell transcriptomes for BALF samples[27], previously deposited in the NCBI's Gene Expression Omnibus database under accession code "GSE145926", and a single-nucleus ASD dataset previously deposited in the NCBI's BioProject database under accession code "PRJNA434002". The list of ligand-receptor interactions employed in our analyses corresponds to the database previously published with CellChat[10], and is available in a "Compendium of Ligand-Receptor Pairs [https://github.com/LewisLabUCSD/Ligand-Receptor-Pairs/blob/master/Human/Human-2020-Jin-LR-pairs.csv]" that we previously published[5]. The data generated in this study for the loadings resulting from the tensor decompositions of the simulated, COVID-19 and ASD datasets are available in the Source Data file. Source data that are not included in this file can be found and reproduced in the Code Ocean capsule. All other relevant data supporting the key findings of this study are available within the article and its Supplementary Information files or from the corresponding author upon reasonable request. Source data are provided with this paper.

## Code availability

All the code used for the analyses in this work is available online in a "Code Ocean capsule [https://doi.org/10.24433/CO.0051950.v2]", which includes the exact version of all tools and software employed, and allows one to perform online a reproducible run of our analyses, outputting pertinent results. Tensor-cell2cell is implemented in our cell2cell suite[61], and its GitHub repository and full documentation can be found at http://lewislab.ucsd.edu/cell2cell/, which also includes comprehensive tutorials that go from raw UMI data to running Tensor-cell2cell, followed by downstream analyses using Tensor-cell2cell's outputs. The code for benchmarking the computational efficiency should be run in a local computer, and is available in a "GitHub repository [https://github.com/LewisLabUCSD/CCC-Benchmark]".

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

## Acknowledgements

E.A. is supported by the Chilean Agencia Nacional de Investigación y Desarrollo (ANID) through its scholarship program DOCTORADO BECAS CHILE/2018—72190270 and by the Fulbright Commission Chile. H.M.B. is supported by NIMH T32GM008806. APL is supported by the InnovaUNAM of the National Autonomous University of Mexico (UNAM) and Alianza UCMX of the University of California. C.A. is supported by NICHD T32HD087978. This work was further supported by NIGMS (R35 GM119850) and the Novo Nordisk Foundation (NNF20SA0066621) to N.E.L.. The authors also thank Daniel McDonald for providing useful guidance about the timing analysis of the tools, the Code Ocean team for providing extra computational time for developing the capsule associated with this work, Aaron Meyer for giving practical insights about tensor decomposition methods, Daniel Dimitrov for providing helpful guidance about running LIANA, and the NVIDIA Academic Hardware Grant Program for supporting the development of Tensor-cell2cell.

## Author contributions

E.A., H.M.B., and N.E.L. conceived the work. C.M. contributed important insights for creating Tensor-cell2cell. E.A. implemented Tensor-cell2cell and performed the analyses on the datasets of COVID-19 and ASD. H.M.B. designed and created the simulated 4D-communication tensor and performed the analyses on the simulated data. E.A., H.M.B., and C.M. performed benchmarking and statistical analyses. C.M. trained classifiers and compared Tensor-cell2cell to CellChat. H.B. performed benchmarking analyses using different external CCC tools. E.A. performed benchmarking analyses using different preprocessing and batch-correction methods. E.A. and H.M.B. developed downstream analyses. A.P.L. helped to interpret the COVID-19 results and researched literature. C.A. helped to interpret the ASD study case and researched literature. R.K. contributed to the benchmarking analyses. E.A. and H.M.B. wrote the paper and all authors carefully reviewed, discussed and edited the paper.

## Competing interests

The authors declare no competing interests.
