## [Peer Review File · Nature Communications]

REVIEWER COMMENTS

Reviewer #1 (Expertise: immunology, COVID19):

In the manuscript 'Context-aware deconvolution of cell-cell communication with Tensor-cell2cell' the authors present a novel method to analyse cell-cell interactions from gene expression data and successfully apply it to a published COVID-19 dataset. While the complexity of 'omics' studies increases, we continue to mostly use matrix-based dimensionality reduction tools. Tensor-cell2cell is novel and would fill a need in the field for analysis methods to match the increased complexity.

I however have one major concern: I can see how Tensor-cell2cell would be superior conceptually, but I fail to see the biological insight superiority of the tool that the authors claim. This is not coming across and should be improved.

- In Fig 3, the authors compare Tensor-cell2cell to CellChat. I understand a direct comparison between tools is not possible due to differences in output. But both tools ultimately lead to a biological conclusion, so can the authors analyse the dataset used in Fig4 with CellChat, and demonstrate what exactly it missed compared to Tensor-cell2cell? Alternatively, the authors provide two references (ref18,Chua et al, ref27, Ren et al) which include ligand-receptor communication analysis of COVID-19 datasets with other tools. Can the authors analyse one of these datasets with Tensor-cell2cell and find something the authors missed?

- In Fig 4, the authors analysed a published COVID-19 dataset of bronchoalveolar lavage fluid (BALF) samples. The published COVID-19 study concludes differences in M1 vs M2 macrophages across disease severity, including expression of CCL2, CCL3 and other markers, while using standard analysis tools. When the authors applied Tensor-cell2cell to their dataset, they reached a similar conclusion. This data in its current format doesn't present Tensor-cell2cell as more insightful, but rather as unnecessary. Could the ligand-receptor interactions identified by Tensor-cell2cell be traced back to specific clusters in the dataset (for instance, as cluster markers), thus identifying specific subpopulations as interacting?

- Is there a sample-size bias in the analysis (other than the obvious very small populations brought up in the discussion)? BALF is usually made up of up to 80% macrophages, and they are by far the biggest population in the dataset used. Therefore, is the epithelial-macrophage interaction detected by Tensor-cell2cell really the most relevant/stronger cell-cell communication or is it a result of cluster size? If there is a bias, this should be stated clearly and discussed. Additionally, analysing a more diverse dataset, such as PBMCs, would be helpful.

Minor comments:

- In the discussion, the authors mention that 'Tensor-cell2cell is not intended for analyzing the behavior of specific pairs of cells': does that mean that the authors cannot focus their analysis on macrophages alone? Ie. instead of setting all immune cells as senders/receivers, use all the different macrophage subclusters, in order to identify the communication of specific macrophage populations.

- Lines 330 and 331: 'our results suggest that COVID-19 severity might be reduced by delivering a drug to control the ratio of M1/M2 macrophages in lungs' should be removed. It is speculative, which does not fit in a results section, and it's also not as much a Tensor-cell2cell observation as an observation made in the original publication.

Reviewer #2 (Expertise: cell-cell communication, OMICS):

Armingol and Baghdassarian et al., presented an efficient and flexible tensor-based method to simultaneously recover CCC patterns (namely ligand-receptor pairs) across multiple contexts. The authors further describe a novel way to generate synthetic, temporal CCC tensor data that was used to support the predictive ability of their method. The authors then showed that their method yields improved classification performance and speed when compared to the context-specific pathway analyses of CellChat. Finally, the authors highlighted that their method is capable of identifying literature supported and clinically relevant CCC interactions in COVID-19 patients. As such, this work occupies a much needed niche for CCC analyses, which if accompanied with some minor extensions would be of particular value to the community.

Major comments

- 1. In this work, the authors presented a comparison to a previously published method applicable in a similar manner to the data. In the text the authors suggest that their method "in most cases, greatly outperformed" (line 355) CellChat. However, this seems to be true only for CellChat's 'structural' analysis, which focuses solely on the signaling network structure, and hence does not consider the similarity of sender and receiver cell types. In contrast, tensor-cell2cell has slightly higher but comparable AUROCs when compared to the 'functional' analysis of CellChat, which should intuitively be the analysis which is more comparable to tensor-cell2cell.**
- 2. On the same note, as the authors also point out, CellChat's analysis focuses on specific pathways, and hence has lower feature space, which possibly puts it at a slight disadvantage in the described performance evaluation. Nevertheless, one of the strengths of cell2cell is the higher dimensionality of its output.**
- 3. Building on this, tensor-cell2cell's output would benefit from some downstream analyses, such as attempting to summarize the output of their tool beyond top ranked interactions, and if possible provide flexible, user-friendly means to do so. This could be achieved via analyses or visualizations (e.g. extensions of the phenotype associations in Fig. 4), such as those performed in CellChat or in tools beyond ligand-receptor analysis such as MOFA.**
- 4. Further, cell2cell was shown to be superior to CellChat's 'structural' analysis, but the advantage of this approach is that it can handle contexts with different cell type composition, which as the authors state would not be appropriate for the current deconvolution algorithm used in tensor-cell2cell. Thus, it only makes sense to show this comparison, if tensor-cell2cell is extended to the deconvolution of heterogeneous cell compositions across contexts.**
- 5. We compliment the authors for their effort in finding literature support for the clinically-relevant COVID-19 ligand-receptor interactions (Supplementary Table S3). However, because one can virtually always find literature support for a given finding, and manual analysis is intrinsically biased, we think that the authors should attempt to also obtain data-driven biological support for the interactions. There are a number of tools which could be useful in this case. For example, in lines 297-299, where the sender role of macrophages is highlighted in regards to a factor associated with the severity of disease. If feasible, to utilize the LR pairs with high loading, which as the authors state are expected to have high variance across contexts, to obtain further biological insights via e.g. pathway enrichment analysis, and hence better explain the signalling events driving the expression of ligands in the macrophages. Alternatively or complementary, one could also complement with other CCC methods (e.g. NicheNet).**

Minor Comments

1. It would be great to show the application of the method to spatially resolved data.
2. In their analyses, the authors use the truncated ligand-receptor means as from the CellPhoneDB paper. It would be particularly interesting to leverage the flexibility of cell2cell and show it using a different scoring function (e.g. CellChat's MM probabilities or NATMI's mean-expression weight), and hence examine how using an alternative scoring function would impact the output of tensor-cell2cell, and if any would change the classifier's performance.
3. The authors further show that their method is more scalable than CellChat particularly when using a GPU, but they do not explicitly mention that CellChat's analysis was not parallelized. This is a rather technical difference between the ease of parallelization in Python when compared to R, and should not be a focus of revisions (as parallelization of CellChat would then lead to a higher RAM usage), but this should be explicitly pointed out.
4. If we imagine ligand-receptor interactions as simple gene sets with 2 genes, then one could suggest that tensor-cell2cell would be extendable to any other gene set given that one could assign a non-negative score between two cell types. Have the authors considered the extension of cell2cell to any 'larger' gene sets? We understand this is out of the scope of this paper, but could be elaborated a bit in the discussion.

Reviewer #3 (Expertise: cell-cell communication, OMICS):

Tensor cell2cell is an interesting method to identify signaling patterns across different datasets/patients/disease stages. The method starts with constructing a 4D cell communication tensor with dimensions context x ligand/receptor pairs x senders x receivers, and then, performs the standard non-negative TCA to find major components of this tensor, interpreting the factor loadings as different "communication patterns". The authors first validated the method with simulation dataset satisfying underlying assumptions and benchmarked the computational efficiency with other methods (mainly CellChat) under multiple contexts situation. As a case study of published dataset, the authors linked the discovered cell communication "factors" using tensor decomposition methods with the covid-19 severeness (control, moderate and severe).

The major novelty of this tensor method is the introduction of "context" dimension and merging it into the tensor decomposition framework. Compared with the pairwise comparison in other methods, this feature enables the more natural and efficient comparison of cell communication among multiple conditions. Other than that this method seems similar to the numerous existing methods, since the rest three dimensions are typically involved in standard cell communication analysis-- especially the non-negative matrix decomposition/factor loading explanation is also used in CellChat, a major benchmarked method repeatedly addressed by authors in this paper.

Currently, many of the methodology and analysis aspect need more explanation and perhaps testing on different types of datasets. Furthermore, the method currently does not provide straightforward ways to visualize the results. Here are some more detailed issues:

Batch effect in the "context" dimension. Introduction of such dimension is the strength of the method, while also brings major concerns about the batch effect, which is prevalent in single-cell datasets. For instance, sometimes the authors directly use "samples" and "patients" as the context conditions where batch effect could be potentially very serious –

do the factor loadings really reflect biological difference or just technical variations? How robust is the method against batch effect, and will data integration techniques improve the analysis? A more rigorous analysis should be conducted on this very important issue.

Benchmarking. While the result is very favorable toward author's method in this paper, the fairness of certain tests is unclear. In terms of efficiency, comparing tensor cell2cell on GPU with other methods on CPU seems not a fair game. And there are multiple factors influencing the computation time and memory management – programming language ,data types used ... not necessarily due to the improvement of author's algorithm/model/method (if so, the authors should explicitly point out the reason and strictly exclude other confounding factors). In terms of accuracy, the authors already acknowledged that the current comparison was indirect since "distinguishing context" is not the main purpose of other methods. Since author's method already provides the novel function of multiple context analysis in cell communication, the current benchmarking of efficiency/accuracy might not be highlighted given the lack of clear standards. Instead, it might be more valuable to directly display the communication results of using cell2cell tensor and Cellchat respectively for real dataset (e.g. Covid-19) and discuss the difference, especially the biological implications.

REVIEWER COMMENTS

Reviewer #1

(Expertise: immunology, COVID19):

In the manuscript 'Context-aware deconvolution of cell-cell communication with Tensor-cell2cell' the authors present a novel method to analyse cell-cell interactions from gene expression data and successfully apply it to a published COVID-19 dataset. While the complexity of 'omics' studies increases, we continue to mostly use matrix-based dimensionality reduction tools. Tensor-cell2cell is novel and would fill a need in the field for analysis methods to match the increased complexity.

I however have one major concern: I can see how Tensor-cell2cell would be superior conceptually, but I fail to see the biological insight superiority of the tool that the authors claim. This is not coming across and should be improved.

We thank the reviewer for their feedback. It is true that multiple results obtained with our tool for the COVID-19 dataset had been previously reported, but most of these results did not come from the study publishing the data. Rather the results surfaced from a number of difficult experiments. Thus, our idea here was to show that Tensor-cell2cell produces biologically meaningful results beyond the primary study. We added a number of clarification points in the Results and Discussions section regarding novel vs previously observed communicatory interactions. We also emphasize that while a number of our results have been observed, many were not observed in the original study that generated the dataset, but rather done so across tens of studies that took many years and resources to generate. In this sense, Tensor-cell2cell can serve as a tool to accelerate researchers' work with their own datasets. Finally, we further analyzed additional datasets, which we have specified below.

- In Fig 3, the authors compare Tensor-cell2cell to CellChat. I understand a direct comparison between tools is not possible due to differences in output. But both tools ultimately lead to a biological conclusion, so can the authors analyze the dataset used in Fig4 with CellChat, and demonstrate what exactly it missed compared to Tensor-cell2cell? Alternatively, the authors provide two references (ref18,Chua et al, ref27, Ren et al) which include ligand-receptor communication analysis of COVID-19 datasets with other tools. Can the authors analyze one of these datasets with Tensor-cell2cell and find something the authors missed?

We thank the reviewer for these suggestions to improve our comparisons with CellChat. We further analyzed the Fig 4 dataset using CellChat and included the outputs from its downstream analyses (specific analyses specified in Supplementary Figures S10-13). We include a discussion comparing and contrasting the biological interpretation of CellChat's and Tensor-

cell2cell's results (Supplementary Notes section *Analysis of the BALF COVID-19 dataset with CellChat*).

We discuss the decreased interpretability of CellChat's output in comparison to Tensor-cell2cell's factors shown in Fig. 4a (Discussion Lines 704-710 in manuscript with suggested edits, or Lines 420-426 in PDF without suggested edits, and Supplementary Notes in the last paragraph of the section *Analysis of the BALF COVID-19 dataset with CellChat*), making biological conclusions using CellChat more difficult. We also demonstrate in an unsupervised manner that Tensor-cell2cell's outputs can lead to better biological conclusions than CellChat's since they separate samples better by COVID-19 severity (Supplementary Figures S3e-h).

With regards to our current analysis of the BALF COVID19 dataset, we added clarification points in the Results, Supplementary Notes, and Discussion sections of new vs previously reported results in the original article of the dataset. Rather than include an additional COVID-19 dataset, we also extend the application of Tensor-cell2cell to an ASD dataset to demonstrate that it can robustly detect communication patterns in differing areas of research. In this section, we focus more on data-driven conclusions from downstream analyses, but again highlight findings that were not reflected in the original publication which generated the data.

- In Fig 4, the authors analysed a published COVID-19 dataset of bronchoalveolar lavage fluid (BALF) samples. The published COVID-19 study concludes differences in M1 vs M2 macrophages across disease severity, including expression of CCL2, CCL3 and other markers, while using standard analysis tools. When the authors applied Tensor-cell2cell to their dataset, they reached a similar conclusion. This data in its current format doesn't present Tensor-cell2cell as more insightful, but rather as unnecessary. Could the ligand-receptor interactions identified by Tensor-cell2cell be traced back to specific clusters in the dataset (for instance, as cluster markers), thus identifying specific subpopulations as interacting?

While the study that generated the dataset detected differences in the population of M1- and M2-like macrophages, they did not link molecular mechanisms as Tensor-cell2cell does, and instead employed markers to identify these populations. Here, by using a communication score that is agnostic to cellular populations, we are still able to detect important CCC patterns discerning severities, which proves that one can analyze datasets without having prior expertise on the topic. In addition, Tensor-cell2cell provides flexibility regarding the input cell type annotations; for example, one can annotate single-cells with different cell-types resolutions, including subpopulations and perform the decomposition, which allows users to trace back subtypes that could be important. We extend Tensor-cell2cell to also use as input the communication scores identified in other tools; some of these communication scoring methods do consider differential expression of ligands and receptors specific to a cell cluster or cell type annotation. We extend the downstream analytical capabilities of Tensor-cell2cell to associate factor-specific LR pairs with sender- and receiver- cell types (using the outer product of loading vectors – see Methods Lines 1060-1079 in manuscript with suggested edits, or Lines 619-628 in

PDF without suggested edits, which may provide information of the interacting subpopulations given their associated markers.

- Is there a sample-size bias in the analysis (other than the obvious very small populations brought up in the discussion)? BALF is usually made up of up to 80% macrophages, and they are by far the biggest population in the dataset used. Therefore, is the epithelial-macrophage interaction detected by Tensor-cell2cell really the most relevant/stronger cell-cell communication or is it a result of cluster size? If there is a bias, this should be stated clearly and discussed. Additionally, analysing a more diverse dataset, such as PBMCs, would be helpful.

As it is currently formatted, Tensor-cell2cell's built-in scoring function is agnostic to cell population because it takes the fraction of single-cells with non zero expression of a given gene to summarize each cell type. In other words, cell frequencies are not considered, and all of them are equally weighted. As such, no bias is introduced as a consequence of cell type composition. If one further wants to consider cell population, Tensor-cell2cell allows other communication scores that may use this information (e.g. CellChat includes an option to specifically weight the scores by cell composition) to be used as input.

Minor comments:

- In the discussion, the authors mention that 'Tensor-cell2cell is not intended for analyzing the behavior of specific pairs of cells': does that mean that the authors cannot focus their analysis on macrophages alone? I.e. instead of setting all immune cells as senders/receivers, use all the different macrophage subclusters, in order to identify the communication of specific macrophage populations.

We thank the reviewer for this comment, as it helped us to improve Tensor-cell2cell by including additional functionalities specified below.

Our intended message with this statement was that although the tensors contain communication scores for every ligand-receptor and cell-cell pair, tensor decomposition is not designed for analyzing one specific combination of pairs of cells and of ligand-receptors at a time, but rather to identify patterns present across combinations of these elements. Instead, to focus on one specific combination, using other tools may be recommended. Nevertheless, we removed this statement to avoid misunderstanding.

Despite the fact that we do not apply Tensor-cell2cell for a specific cell and LR pair combination, we specify a couple caveats: 1) we add a function to subset the tensor to cell types of interest (as well as other tensor dimensions) to further focus the analysis. In this sense, if there are multiple macrophage subpopulations present, the tensor can be subsetted to those subpopulations to identify macrophage-specific communication or can be subsetted in combination with any other cell types one thinks may play an important role in macrophage communication. 2) If an output factor shows distinct communication patterns for a particular cell

type that one wants to further dissect, we've included a number of downstream factor-specific analyses for this purpose (Fig. 5, Fig. S9).

- Lines 330 and 331: 'our results suggest that COVID-19 severity might be reduced by delivering a drug to control the ratio of M1/M2 macrophages in lungs' should be removed. It is speculative, which does not fit in a results section, and it's also not as much a Tensor-cell2cell observation as an observation made in the original publication. Thanks for this observation, we have now removed this sentence from the manuscript.

Reviewer #2

(Expertise: cell-cell communication, OMICS):

Armingol and Baghdassarian et al., presented an efficient and flexible tensor-based method to simultaneously recover CCC patterns (namely ligand-receptor pairs) across multiple contexts. The authors further describe a novel way to generate synthetic, temporal CCC tensor data that was used to support the predictive ability of their method. The authors then showed that their method yields improved classification performance and speed when compared to the context-specific pathway analyses of CellChat. Finally, the authors highlighted that their method is capable of identifying literature supported and clinically relevant CCC interactions in COVID-19 patients. As such, this work occupies a much needed niche for CCC analyses, which if accompanied with some minor extensions would be of particular value to the community.

Major comments

1. In this work, the authors presented a comparison to a previously published method applicable in a similar manner to the data. In the text the authors suggest that their method "in most cases, greatly outperformed" (line 355) CellChat. However, this seems to be true only for CellChat's 'structural' analysis, which focuses solely on the signaling network structure, and hence does not consider the similarity of sender and receiver cell types. In contrast, tensor-cell2cell has slightly higher but comparable AUROCs when compared to the 'functional' analysis of CellChat, which should intuitively be the analysis which is more comparable to tensor-cell2cell.

We thank the reviewer for their insights to CellChat's pipeline and whether it may be appropriately compared with Tensor-cell2cell. Firstly, we agree that the comparison between the two tools glosses over the fact that the two methods are quite different. As such, while the improved performance holds true, we de-emphasize this portion of the analysis by moving it to the supplement, and replace it with a demonstration of how Tensor-cell2cell is complementary to many other tools by serving as a method to further process and analyze the output of many other tools. Thus, we show Tensor-cell2cell can extend the types of analyses CellChat and other communication scoring tools. In addition, we clarified this in the Supplementary Notes. Please note our response to your comment #4 for additional relevant discussion.

2. On the same note, as the authors also point out, CellChat's analysis focuses on specific pathways, and hence has lower feature space, which possibly puts it at a slight disadvantage in the described performance evaluation. Nevertheless, one of the strengths of cell2cell is the higher dimensionality of its output.

We thank the reviewer for pointing out the advantages of Tensor-cell2cell outputs versus CellChat's, and indeed linking higher dimensions could enable better conclusions. We believe that to make fair comparisons between our tool and CellChat, the best option was to train the classifiers with loadings from the sample dimensions from the decomposition, and we did not use extra information such as LR pair- or sender/receiver-cell-associated loadings. Thus, we felt that using the UMAP values that CellChat generates for each sample is comparable to using the sample loadings from Tensor-cell2cell. Nonetheless, we agree that the comparison does not fully account for the differences of capabilities between the tools, so we have de-emphasized this analysis altogether. Furthermore, since we extended Tensor-cell2cell's functionality to use other methods' communication scores as input, this can include CellChat's pathway level scores.

3. Building on this, tensor-cell2cell's output would benefit from some downstream analyses, such as attempting to summarize the output of their tool beyond top ranked interactions, and if possible provide flexible, user-friendly means to do so. This could be achieved via analyses or visualizations (e.g. extensions of the phenotype associations in Fig. 4), such as those performed in CellChat or in tools beyond ligand-receptor analysis such as MOFA.

We thank the reviewer for this suggestion, and believe that all the reviewers' recommendations to add more downstream analyses has greatly improved the utility of our tool. We have expanded and clarified the downstream analytical capabilities of Tensor-cell2cell. This includes 1) statistical, 2) correlative, and 3) clustering analyses to understand context associations for each factor, 4) gene set enrichment analysis of ligand-receptor loadings to identify granular signaling pathways associated with factors, 5) the generation of factor-specific cell-cell communication networks to represent the overall communication state of cells in that factor, and 6) using this factor-specific network to understand the communication distribution inequality between sender- and receiver- cells. We focused on these downstream capabilities in our new ASD analysis (Fig. 5) and highlighted them in a separate Methods section.

4. Further, cell2cell was shown to be superior to CellChat's 'structural' analysis, but the advantage of this approach is that it can handle contexts with different cell type composition, which as the authors state would not be appropriate for the current deconvolution algorithm used in tensor-cell2cell. Thus, it only makes sense to show this comparison, if tensor-cell2cell is extended to the deconvolution of heterogeneous cell compositions across contexts.

Although we mention that the decomposition algorithm can be better optimized for handling missing values (i.e., NaN communication scores), through discussions with Tensorly

developers, we understand that the native non-negative PARAFAC algorithm used by Tensor-cell2cell can handle missing values. Furthermore, the decomposition algorithm can already handle communication scores of zero. As such, Tensor-cell2cell allows to consider the union or intersection of both the ligand-receptor and cell type dimensions across samples (parameter “how=’inner” for using the intersection or “how=’outer” for using the union when building the tensor); when taking the union, our tool automatically deals with missing values by masking them as zeros. Thus, Tensor-cell2cell should still be comparable to the CellChat “structural” analysis.

Finally, the comparison with the “structural” analysis of CellChat was performed in similar conditions to Tensor-cell2cell, without using cell types that were not present in all conditions. Thus, we are only measuring their computational efficiency to handle the same elements, and distinguish how computationally intensive they could be.

5. We compliment the authors for their effort in finding literature support for the clinically-relevant COVID-19 ligand-receptor interactions (Supplementary Table S3). However, because one can virtually always find literature support for a given finding, and manual analysis is intrinsically biased, we think that the authors should attempt to also obtain data-driven biological support for the interactions. There are a number of tools which could be useful in this case. For example, in lines 297-299, where the sender role of macrophages is highlighted in regards to a factor associated with the severity of disease. If feasible, to utilize the LR pairs with high loading, which as the authors state are expected to have high variance across contexts, to obtain further biological insights via e.g. pathway enrichment analysis, and hence better explain the signalling events driving the expression of ligands in the macrophages. Alternatively or complementary, one could also complement with other CCC methods (e.g. NicheNet).

We thank the reviewer for these suggestions. We chose to retain the COVID-19 analysis as is, and instead added a new analysis of an ASD dataset using more downstream analyses to assess the decomposition outputs. Our intention here was to demonstrate that researchers who may use Tensor-cell2cell on their own datasets can take either a more manual approach of assessing the factors, as with the COVID analysis, or a more data-driven approach as with the ASD analysis.

As indicated by the reviewer, one can use the LR loadings for further analysis. For that reason, we also thank the reviewer for this insightful comment. We preranked LR pairs by their loadings and used them as inputs for analyzing enriched pathways through GSEA. For this purpose we also built LR pathway sets from KEGG as indicated in the Methods.

Minor Comments

1. It would be great to show the application of the method to spatially resolved data.

This is an important application that we would have liked to further expand. However, the current state of spatial transcriptomics is very shallow, which strongly limits the number of LR

pairs we could evaluate through tensor decomposition, leading to obtaining not clear general patterns. We included another dataset analysis, but instead it was about ASD.

2. In their analyses, the authors use the truncated ligand-receptor means as from the CellPhoneDB paper. It would be particularly interesting to leverage the flexibility of cell2cell and show it using a different scoring function(e.g. CellChat's MM probabilities or NATMI's mean-expression weight), and hence examine how using an alternative scoring function would impact the output of tensor-cell2cell, and if any would change the classifier's performance.

We extended Tensor-cell2cell to build 4D-Communication Tensors from the output of other communication scoring tools. We use the CorrIndex, rather than the classifier, to assess the consistency of decomposition outputs (Fig 3a). We again thank the author for this comment, as we believe it results in major improvements to the utility of Tensor-cell2cell.

3. The authors further show that their method is more scalable than CellChat particularly when using a GPU, but they do not explicitly mention that CellChat's analysis was not parallelized. This is a rather technical difference between the ease of parallelization in Python when compared to R, and should not be a focus of revisions (as parallelization of CellChat would then lead to a higher RAM usage), but this should be explicitly pointed out.

This is an important point that we believe we were unclear about. We clarified these points by mentioning that our analysis compared Tensor-cell2cell and CellChat without parallelization, and as an extra feature added the GPU evaluation to show the final user that they can further accelerate the analysis. Finally, we now explicitly point out that some differences could be due to the use of Python vs R.

4. If we imagine ligand-receptor interactions as simple gene sets with 2 genes, then one could suggest that tensor-cell2cell would be extendable to any other gene set given that one could assign a non-negative score between two cell types. Have the authors considered the extension of cell2cell to any 'larger' gene sets? We understand this is out of the scope of this paper, but could be elaborated a bit in the discussion.

The tensor formulation does make it flexible for different input types, and it would be interesting to extend what is currently the ligand-receptor dimension to other information encoded by the gene expression matrix that links two cell types. We agree that this could extend the power of decomposition in identifying communication patterns and have specified this in the Discussion Lines 665-668 in manuscript with suggested edits, or Lines 408-411 in PDF without suggested edits.

Reviewer #3

(Expertise: cell-cell communication, OMICS):

Tensor cell2cell is an interesting method to identify signaling patterns across different datasets/patients/disease stages. The method starts with constructing a 4D cell communication tensor with dimensions context x ligand/receptor pairs x senders x receivers, and then, performs the standard non-negative TCA to find major components of this tensor, interpreting the factor loadings as different “communication patterns”. The authors first validated the method with simulation dataset satisfying underlying assumptions and benchmarked the computational efficiency with other methods (mainly CellChat) under multiple contexts situation. As a case study of published dataset, the authors linked the discovered cell communication “factors” using tensor decomposition methods with the covid-19 severeness (control, moderate and severe).

The major novelty of this tensor method is the introduction of “context” dimension and merging it into the tensor decomposition framework. Compared with the pairwise comparison in other methods, this feature enables the more natural and efficient comparison of cell communication among multiple conditions .Other than that this method seems similar to the numerous existing methods, since the rest three dimensions are typically involved in standard cell communication analysis-- especially the non-negative matrix decomposition/factor loading explanation is also used in CellChat, a major benchmarked method repeatedly addressed by authors in this paper.

While other tools do consider the sender-, receiver-, and LR- dimensions included in our 4D-Communication Tensor, the key discrepancy is how the data is structured (other approaches arrange elements of these dimensions as matrices, missing important data-relationships). The major benefit of structuring it in the tensor format that includes the context-dimension is that the decomposition considers the relationships between all 4 dimensions simultaneously to extract relevant communication patterns, i.e. combinations of elements in each dimension. As we mention in the Introduction, the tensor structure allows for more robust decompositions as compared to matrix decompositions – specifically, there are less stringent requirements to arrive at a unique solution and inherent relationships between tensor slices in the low-rank approximation (<https://arxiv.org/pdf/1711.10781.pdf>).

In the case of CellChat, even though it does consider senders, receivers and LR pairs, this tool however performs two independent NMFs, one on a 2D-matrix with sender cells and LR pairs and the other on a 2D-matrix with receiver cells and LR pairs, then the patterns found in each case are connected to have an idea of the links between sender-receiver cells. This two-steps analysis misses the natural interrelationships between the three dimensions together, that would be captured with a tensor decomposition instead of NMF, and altogether disregards context.

Finally, we acknowledge the existence of one tool, scTensor (<https://doi.org/10.1101/566182>), that uses a 3D-tensor (without the context dimension), and performs a tensor decomposition. However, the main difference is the algorithm chosen. scTensor uses a Tucker decomposition, which offers different interpretations and conclusions than the CANDECOMP/PARAFAC (CP) decomposition, the algorithm used by Tensor-cell2cell. The CP algorithm has advantages of

interpretability over the Tucker decomposition, and more direct application of the loadings for factor-wise downstream analysis.

Currently, many of the methodology and analysis aspects need more explanation and perhaps testing on different types of datasets. Furthermore, the method currently does not provide straightforward ways to visualize the results. Here are some more detailed issues:

We have added clarifications to our methodology and analysis throughout the manuscript, and we have included an additional ASD dataset we analyzed. We have also included a number of downstream analyses with options to visualize the results. These analyses include 1) statistical, 2) correlative, and 3) clustering analyses to understand context associations for each factor, 4) gene set enrichment analysis of ligand-receptor loadings to identify granular signaling pathways associated with factors, 5) the generation of factor-specific cell-cell communication networks to represent the overall communication state of cells in that factor, and 6) using this factor-specific network to understand the communication distribution inequality between sender- and receiver-cells. We focused on these downstream capabilities in our new ASD analysis (Fig. 5, Supplementary Figure S9) and highlighted them in a separate Methods section.

Batch effect in the “context” dimension. Introduction of such dimension is the strength of the method, while also brings major concerns about the batch effect, which is prevalent in single-cell datasets. For instance, sometimes the authors directly use “samples” and “patients” as the context conditions where batch effect could be potentially very serious – do the factor loadings really reflect biological difference or just technical variations? How robust is the method against batch effect, and will data integration techniques improve the analysis? A more rigorous analysis should be conducted on this very important issue.

We thank the reviewer for their suggestion to account for how batch effects may skew the results. We agree that the technical variance this can introduce may be detrimental to the interpretation of results when implementing such tools that analyze multiple samples. As such, we include a new analysis to assess the robustness of our decomposition results to varying preprocessing and batch-correction methods. We find that, with the exception of applying no transformation (i.e., using the raw counts), the decomposition tends to be consistent across methods (Fig 3b). Specifically, we used the CorrIndex metric to assess consistency of decomposition outputs on 5 different gene expression preprocessing procedures – raw counts, $\log(1+CPM)$, fraction of non-zero cells, Scanorama batch correction, and ComBat batch correction. We also clarify in the Discussion (Lines 697-699 in manuscript with suggested edits, or Lines 413-415 in PDF without suggested edits) that appropriate batch correction steps should be taken to mitigate this effect.

Given this, we retain samples in the context-dimension to achieve the best resolution possible. We also would like to mention that if batch effects were strong and decompositions reflected

sample-specific rather than biological differences, we would expect decomposition factors that reflect a high loading of a single sample.

Benchmarking. While the result is very favorable toward author’s method in this paper, the fairness of certain tests is unclear. In terms of efficiency, comparing tensor cell2cell on GPU with other methods on CPU seems not a fair game.

We agree that including GPU and CPU comparisons of efficiency would have been biased if we had used the methods on different hardware and setups. For that reason we have now clarified that we performed a fair comparison by running both tools, Tensor-cell2cell and CellChat, on a CPU without parallelization; and indicated that the GPU evaluation is a demonstrative case to show users how much they could accelerate their analysis if they have an available GPU. In other words, we highlight GPU as an additional benefit that our tool offers rather than a direct comparison. Furthermore, we decided to de-emphasize the benchmarking analysis by moving most of it to the Supplementary Notes. We choose to replace this analysis with one showing how Tensor-cell2cell may build off of existing methods, framing it as an extension to tools such as CellChat.

And there are multiple factors influencing the computation time and memory management – programming language ,data types used ... not necessarily due to the improvement of author’s algorithm/model/method (if so, the authors should explicitly point out the reason and strictly exclude other confounding factors).

We acknowledge that computational efficiency could be affected by other factors. Indeed, the goal of our runtime benchmarking was not to explicitly compare the algorithmic performance (which is a topic previously evaluated in the area of tensor decompositions, as for example: <https://doi.org/10.48550/arXiv.1506.04209>; <https://doi.org/10.1016/j.cam.2021.113972>), and instead we focus on the benefits the final users would have when running our tool on their machines. For that reason, we also clarified in the benchmarking section (Supplementary Notes) that differences of performance could be due to the use of different programming languages.

In terms of accuracy, the authors already acknowledged that the current comparison was indirect since “distinguishing context” is not the main purpose of other methods. Since author’s method already provides the novel function of multiple context analysis in cell communication, the current benchmarking of efficiency/accuracy might not be highlighted given the lack of clear standards. Instead, it might be more valuable to directly display the communication results of using cell2cell tensor and Cellchat respectively for real dataset (e.g. Covid-19) and discuss the difference, especially the biological implications.

We thank the reviewer for this suggestion. We actually decided to directly use CellChat’s raw communication scores, along with a few other methods’ scores, to build and decompose 4D-Communication Tensors. We find that the decomposition results across communication methods seem to be consistent (Fig. 3a and Supplementary Figure S4).

We also further analyzed the Fig 4 dataset using CellChat and included the outputs from its downstream analyses (specific analyses specified in Supplementary Figures S10-13). We include a discussion comparing and contrasting the biological interpretation of CellChat's and Tensor-cell2cell's results (Supplementary Notes section *Analysis of the BALF COVID-19 dataset with CellChat*).

We discuss the decreased interpretability of CellChat's output in comparison to Tensor-cell2cell's factors shown in Fig. 4a (Discussion Lines 704-710 in manuscript with suggested edits, or Lines 420-426 in PDF without suggested edits, and Supplementary Notes in the last paragraph of the section *Analysis of the BALF COVID-19 dataset with CellChat*), making biological conclusions using CellChat more difficult. We also demonstrate in an unsupervised manner that Tensor-cell2cell's outputs can lead to better biological conclusions than CellChat's since they separate samples better by COVID-19 severity (Supplementary Figures S3e-h).

REVIEWER COMMENTS

Reviewer #1 (Remarks to the Author):

The authors have provided edits to the manuscript, improved the benchmarking of their tool, and most importantly, added an additional dataset that helps support the impact of Tensor-cell2cell. These revisions address my concerns. I believe this research article will make a worthy and useful contribution to the field.

Reviewer #2 (Remarks to the Author):

In general, the authors have addressed most of the concerns raised by us (and, in our opinion, from the other reviewers). In particular, they have removed exaggerations and de-emphasized the comparison to CellChat's analysis. More importantly, they extended the methods' utilities to a number of valuable data-driven downstream analyses, and have shown the robustness of their method to alternative ligand-receptor scoring functions and different batch-effect correction techniques.

That being said, before the manuscript is accepted for publication, we want to stress the following point:

Major comment:

As previously suggested, we firmly believe that the value of a method is directly reflected by its usability and subsequently its value to generate unbiased hypotheses. We compliment the authors for addressing this point in the manuscript, but this is yet to be addressed for the case of the future average user of their method.

To this point, we expect the authors to provide further documentation for their method and a truly user-friendly, start-to-finish (processed data -> downstream analyses) tutorial. A tutorial which comprehensively describes the outputs and analyses presented in the ASD analyses would be of particular value, as the currently provided tutorial (https://github.com/earmingol/cell2cell/blob/master/examples/tensor_cell2cell/Tensor-cell2cell-PBMC.ipynb) is simply insufficient to highlight the value and possible applications of their method.

We also want to underline that for the average user, a gini coefficient of decomposed intercellular communication tensors, among other similar phrases, is a distant and extremely abstract idea. So, we kindly ask the authors to attempt to provide some 'biological' guidance (in the said tutorial) of how one could interpret the outputs of their method.

We again refer the authors to examples of comprehensive tutorials and documentation of tools such as MOFA (<https://biofam.github.io/MOFA2/tutorials.html>), Scanpy (<https://scanpy.readthedocs.io/en/stable/>), Scirpy (<https://scverse.org/scirpy/latest/>), NicheNet, and CellChat, among many others.

Minor comments:

Is there any reason why there are interactions between duplicated proteins in Supplementary Table S4? For example, NEGR1-NEGR1 and ICAM1-ICAM1.

The authors should consider changing "disease condition" to condition or category in Figure 5's legend.

Reviewer #3 (Remarks to the Author):

The revision is well done. No more comments.

Reviewer #1

The authors have provided edits to the manuscript, improved the benchmarking of their tool, and most importantly, added an additional dataset that helps support the impact of Tensor-cell2cell. These revisions address my concerns. I believe this research article will make a worthy and useful contribution to the field.

We thank the reviewer for their feedback and positive comments about our method. We hope it will be useful for the community.

Reviewer #2

In general, the authors have addressed most of the concerns raised by us (and, in our opinion, from the other reviewers). In particular, they have removed exaggerations and de-emphasized the comparison to CellChat's analysis. More importantly, they extended the methods' utilities to a number of valuable data-driven downstream analyses, and have shown the robustness of their method to alternative ligand-receptor scoring functions and different batch-effect correction techniques.

That being said, before the manuscript is accepted for publication, we want to stress the following point:

Major comment:

As previously suggested, we firmly believe that the value of a method is directly reflected by its usability and subsequently its value to generate unbiased hypotheses. We compliment the authors for addressing this point in the manuscript, but this is yet to be addressed for the case of the future average user of their method.

To this point, we expect the authors to provide further documentation for their method and a truly user-friendly, start-to-finish (processed data -> downstream analyses) tutorial. A tutorial which comprehensively describes the outputs and analyses presented in the ASD analyses would be of particular value, as the currently provided tutorial (https://github.com/earmingol/cell2cell/blob/master/examples/tensor_cell2cell/Tensor-cell2cell-PBMC.ipynb) is simply insufficient to highlight the value and possible applications of their method.

We also want to underline that for the average user, a gini coefficient of decomposed intercellular communication tensors, among other similar phrases, is a distant and extremely abstract idea. So, we kindly ask the authors to attempt to provide some

‘biological’ guidance (in the said tutorial) of how one could interpret the outputs of their method.

We again refer the authors to examples of comprehensive tutorials and documentation of tools such as MOFA (<https://biofam.github.io/MOFA2/tutorials.html>), Scanpy (<https://scanpy.readthedocs.io/en/stable/>), Scirpy (<https://scverse.org/scirpy/latest/>), NicheNet, and CellChat, among many others.

We thank the reviewer for their comments. We are much in agreement about the user-friendly tutorial and the biological guidance. Given its importance, we have created a webpage for Tensor-cell2cell (and its umbrella tool, cell2cell) which contains full documentation of the tool and multiple tutorials (including previously existing ones and new ones). It can be found here: <https://earmingol.github.io/cell2cell/>

On this webpage, we have full API documentation of the tool’s functions following the Python community standard numpy docstring style. This delineates the exact inputs and their format. This documentation can be found here: <https://earmingol.github.io/cell2cell/documentation/>.

Furthermore, we have clarified the location of all our various tutorials and codebases for existing Tensor-cell2cell analyses here: <https://earmingol.github.io/cell2cell/#examples>

Finally, we have created additional tutorials, built around the ASD dataset, that show the user how to run a comprehensive analysis starting from raw UMI counts. This includes:

1) data preprocessing and running the decomposition algorithm, which allows the user to obtain the loadings which reveal the communication patterns. Available here:

<https://earmingol.github.io/cell2cell/tutorials/ASD/01-Tensor-Factorization-ASD/>

2) downstream analyses that clearly state the biological interpretation of each analysis. These analyses include a section of context group comparisons (box plots), factor-specific communication networks, extensive explanation of the use and interpretation of Gini coefficients, and clustering of samples/contexts and key ligand-receptor pairs. Available here:

<https://earmingol.github.io/cell2cell/tutorials/ASD/02-Factor-Specific-ASD/>

3) A full tutorial on how to generate LR sets that are used for running GSEA on the loadings of the LR interactions ranked by their factor-specific loadings, and how to generate the interpretable figures. Available here: <https://earmingol.github.io/cell2cell/tutorials/ASD/03-GSEA-ASD/>

We believe the downstream analyses tutorials strongly complement our existing explanations of the biological interpretation of the ASD dataset in the results section of our manuscript (those associated with Figure 5 and Supplementary Figure S9).

Minor comments:

Is there any reason why there are interactions between duplicated proteins in Supplementary Table S4? For example, NEGR1-NEGR1 and ICAM1-ICAM1.

Ligand-receptor interaction between the same protein can be found in PPI databases and in real-world biology. They are common for cell contacts (e.g. NEGR1 is a neural cell adhesion molecule). The specific interactions are dependent on the database that one uses as input of Tensor-cell2cell. In our case, these interactions are present in CellChat's database. Since they are expressed in both the sender and receiver cells, and seem to be important for the corresponding factors, Tensor-cell2cell assigned them high loadings.

The authors should consider changing “disease condition” to condition or category in Figure 5’s legend.

Thank you for pointing it out. We have now corrected this in the manuscript.

Reviewer #3

The revision is well done. No more comments.

We thank the reviewer for their feedback.

REVIEWERS' COMMENTS

Reviewer #2 (Remarks to the Author):

We are fully satisfied with the manuscript and commend the authors for their excellent work.

Reviewer #4 (Remarks to the Author):

We thank the authors for addressing our final point, which they have done fully. We have no further comments and congratulate the authors for their excellent work.

REVIEWERS' COMMENTS

Reviewer #2 (Remarks to the Author):

We are fully satisfied with the manuscript and commend the authors for their excellent work.

Reviewer #4 (Remarks to the Author):

We thank the authors for addressing our final point, which they have done fully. We have no further comments and congratulate the authors for their excellent work.

We thank the reviewers for the feedback provided through the peer-review process. We think that their comments considerably helped to improve the manuscript and the contribution it could make to the field.